# Typical and atypical language brain organization based on intrinsic connectivity and multitask functional asymmetries

Loïc Labache[1,2,3,4,5,6], Bernard Mazoyer[1,2,3,7]*, Marc Joliot[1,2,3], Fabrice Crivello[1,2,3], Isabelle Hesling[1,2,3], Nathalie Tzourio-Mazoyer[1,2,3]

[1]Université de Bordeaux, Institut des Maladies Neurodégénératives, UMR 5293, Groupe d'Imagerie Neurofonctionnelle, Bordeaux, France; [2]CNRS, Institut des Maladies Neurodégénératives, UMR 5293, Groupe d'Imagerie Neurofonctionnelle, Bordeaux, France; [3]CEA, Institut des Maladies Neurodégénératives, UMR 5293, Groupe d'Imagerie Neurofonctionnelle, Bordeaux, France; [4]Université de Bordeaux, Institut de Mathématiques de Bordeaux, UMR 5251, Bordeaux, France; [5]Bordeaux INP, Institut de Mathématiques de Bordeaux, UMR 5251, Bordeaux, France; [6]INRIA Bordeaux Sud-Ouest, Institut de Mathématiques de Bordeaux, UMR 5251, Contrôle de Qualité et Fiabilité Dynamique, Talence, France; [7]Centre Hospitalier Universitaire de Bordeaux, Bordeaux, France

**Abstract** Based on the joint investigation in 287 healthy volunteers (150 left-Handers (LH)) of language task-induced asymmetries and intrinsic connectivity strength of the sentence-processing supramodal network, we show that individuals with atypical rightward language lateralization (N = 30, 25 LH) do not rely on an organization that simply mirrors that of typical leftward lateralized individuals. Actually, the resting-state organization in the atypicals showed that their sentence processing was underpinned by left and right networks both wired for language processing and highly interacting by strong interhemispheric intrinsic connectivity and larger corpus callosum volume. Such a loose hemispheric specialization for language permits the hosting of language in either the left and/or right hemisphere as assessed by a very high incidence of dissociations across various language task-induced asymmetries in this group.

*For correspondence:
bernard.mazoyer@u-bordeaux.fr

Competing interests: The authors declare that no competing interests exist.

## Introduction

Hemispheric specialization, and more particularly hemispheric specialization for language, can be defined as *"... a hemisphere-dependent relationship between a cognitive, sensory, or motor function and a set of brain structures. It includes both the hosting by a given hemisphere of specialized networks that have unique functional properties and mechanisms that enable the inter-hemispheric coordination necessary for efficient processing"* (*Hervé et al., 2013*). Major issues on the topic of hemispheric functional segregation have been listed in a recent review article (*Vingerhoets, 2019*). Highlighting the importance of in-depth investigations of individuals exhibiting atypical hemispheric lateralization for language production, several burning questions related to this atypical phenotype were identified including the characterization of its regional pattern, its relationship with handedness, its structural underpinnings, whether such atypicality holds for other cognitive functional phenotypes, and whether it is associated with variations in behavior and/or cognitive abilities.

To comprehensively understand typical and atypical hemispheric organization for high-order language processing, it is necessary to examine the functional organization of the language network in

the dominant hemisphere together with its interhemispheric coordination with the mirroring network in the opposite hemisphere. Such an approach can be completed by integrating different and complementary neuroimaging information provided by resting-state and task-induced activation investigations.

Task-induced functional asymmetries are reliable markers for assessing individuals' hemispheric specialization for language, as attested by the very good concordance between fMRI asymmetries measured during language tasks and Wada testing, which is considered the gold standard to measure dominance (*Dym et al., 2011*). This methodology makes it possible to revisit the incidence of atypical organization in healthy individuals in relationship with handedness. The earliest research on this topic reported that rare individuals presenting a shift in lateralization—having rightward asymmetry during language tasks—can be found in left-handers (*Hund-Georgiadis et al., 2002*; *Pujol et al., 1999*; *Szaflarski et al., 2002*). This last observation is consistent with previous aphasia studies (*Hécaen and Sauguet, 1971*) and investigations of epileptic patients with Wada testing (*Isaacs et al., 2006*), but the very low incidence of atypical individuals coupled with the low incidence of left handedness is a difficulty in the assessment of language lateralization variability in healthy individuals. To overcome this issue, we gathered a database of healthy volunteers specifically enriched in left-handers (BIL and GIN; *Mazoyer et al., 2016*) and measured the Hemispheric Functional Lateralization index (HFLI; *Wilke and Schmithorst, 2006*) for sentence production in 297 of its participants. We thereby uncovered 3 patterns of language lateralization, namely, 10 strong-atypical individuals with strong rightward lateralization that included only left-handers, 37 ambilateral individuals, including 23 left-handers, with weak or no lateralization and 250 typical individuals with strong leftward lateralization, including 120 left-handers (*Mazoyer et al., 2014*). In a subsequent investigation of regional asymmetries in these same participants, we provided evidence that there were no differences between typical right- and left-handers in terms of regional patterns of asymmetry. In contrast, left-handed ambilaterals were not lateralized unlike right-handed ambilaterals who showed a modest leftward asymmetry (*Tzourio-Mazoyer et al., 2016*). Left-handed ambilaterals were also characterized by higher connectivity at rest across homotopic language regions suggesting that enhanced interhemispheric cooperation at rest is a marker of increased interhemispheric cooperation associated with decreased asymmetries during sentence minus list production contrast (*Tzourio-Mazoyer et al., 2016*).

This last result highlights the importance of resting-state fMRI for the investigation of language network organization, as it makes it possible to measure the functional intrinsic connectivity of networks within each hemisphere and the differences in connectivity between the hemispheres. *Raemaekers et al., 2018* showed a significant association across individuals between the asymmetry in functional connectivity scores and the asymmetries in language task lateralization scores measured in language regions located along the longitudinal fissure in 423 healthy volunteers. Consistent with the Raemaekers et al. report, we have recently shown that in the left hemisphere, the resting-state degree centrality (Rs_DC), or strength of connectivity, was significantly correlated with task-induced activations in the supramodal network of regions dedicated to sentence processing (SENT_CORE; *Labache et al., 2019*). Moreover, the Reynolds' study involving 117 children demonstrated that asymmetry in the strength of connectivity between language areas followed the same developmental pattern of increases in asymmetry between 2 and 7 years old (*Reynolds et al., 2019*) as that reported with task-induced activations (*Friederici et al., 2011*; *Perani et al., 2011*). Finally, such asymmetries in within-hemisphere intrinsic connectivity at rest are modified in individuals with rightward lateralization for language production (*Joliot et al., 2016*). Taken together, these studies demonstrate that language network intrinsic connectivity and its asymmetry are important markers to characterize the variability in language organization in the brain.

A key issue that remains unresolved regarding the typical and atypical organization for language in healthy individuals is that of dissociation. Actually, there is very little knowledge on homogeneity in lateralization across different language components in healthy individuals. Neuropsychological studies such as the seminal study conducted by Hécaen in left-handers (*Hécaen and Sauguet, 1971*; *Hécaen et al., 1981*) have shown that hemispheric dominance is not a property of a given hemisphere but rather that the dominant hemisphere may shift for different language functions in some individuals. After left hemisphere lesions, left-handed aphasic patients can show rare deficits in comprehension, while deficits in production are constant, indicating a dissociation of these two language components (*Hécaen et al., 1981*). PET studies with healthy volunteers have provided evidence of

dissociations between production and comprehension in rare left-handers, with a leftward asymmetry during production and a rightward asymmetry during comprehension (for example, *Tzourio-Mazoyer et al., 2004*). In pathological states, particularly epilepsy, several studies have reported dissociations between asymmetries for language production and those for listening (*Baciu et al., 2003*; *Kurthen et al., 1994*; *Kurthen et al., 1992*; *Lee et al., 2008*), and the results of a longitudinal study of Wada testing in four of these patients suggested that language production was more likely to shift hemispheres than speech comprehension (*Lee et al., 2008*).

The occurrence of dissociation, particularly in some healthy individuals, suggests a potential independence of different language components in terms of hemispheric dominance, which calls for the search for factors in brain organization that could allow different language components to be hosted in different hemispheres. One hypothesis could be that a loose organization in terms of lateralization, marked by some bilateral involvement of language areas and strong interhemispheric connectivity, would make it possible for different language components to be lateralized in different hemispheres. In such a case, one should observe an increased occurrence of dissociations in atypicals, and a better knowledge of the characteristics of the individuals who are more likely to host dissociations will make it possible to optimize the neuroimaging paradigm used to determine language lateralization. For example, as implemented in presurgical evaluations of epileptic patients (*Baciu et al., 2003*; *Baciu et al., 2005*), a paradigm that includes a battery of language tasks in addition to production could be designed for this subpopulation.

Regarding these matters, the present study, which includes measures of lateralization during production, listening and reading tasks in the same participants, is an opportunity to refine the understanding of dissociations. Reading is the last language function acquired through development, since the emergence of this language component relies on strong interactions between speech, eye motor systems, and preorthographic processing by visuospatial attentional areas whose lateralization is located in different hemispheres (*Lobier et al., 2012*; *Petit et al., 2015*). One might suggest that dissociations between the lateralization of speech comprehension and production could occur from two different sources of variability: speech perception (*Zatorre et al., 2002*) and motor control of speech (*Lieberman et al., 2007*), respectively. Nevertheless, examining reading lateralization, which is established later on the basis of comprehension and production lateralization, will allow us to enlarge the question of the possible sources of interindividual variability in language lateralization to that of the relationships between rightward lateralized visuospatial functions and leftward lateralized language functions.

Hemispheric asymmetries in gray and white matter have been used to investigate variability in hemispheric specialization for language, although these measures mainly provide information on inherited gross anatomical differences between the two hemispheres, which are observable at the whole brain level as a global torsion of the brain (i.e. the Yakovlevian torque; *Toga and Thompson, 2003*). In areas related to language processing, such as the *planum temporale* close to the sylvian fissure, leftward asymmetries of fissure depth are seen in utero (*Habas et al., 2012*), and these asymmetries are of the same amplitude at birth as in adults (*Hill et al., 2010*), showing no subsequent modifications during development (*Li et al., 2014*). Notably, in adults, the gross leftward asymmetry of the *planum temporale* does not have the characteristics of a marker of hemispheric dominance at the individual level (*Tzourio-Mazoyer et al., 2018*). Even if some local relationships were found between gray matter and language task-induced functional asymmetries during word-listening, they explain only a small fraction of the interindividual variability of local functional asymmetries (*Josse et al., 2009*). The *corpus callosum,* made of fibers connecting both hemispheres, has also been investigated as it is the main anatomical support for interhemispheric connectivity. The *corpus callosum* surface or volume has thus been measured as a potential anatomical marker of this interhemispheric connectivity. Actually, during the course of phylogenesis, increasing brain volumes go along with decreasing corpus callosum volumes relative to brain size (review in *Hopkins and Cantalupo, 2008*). On this basis, one should expect that a strong lateralization would be associated with a smaller corpus callosum volume, as previously observed in males for anatomical hemispheric asymmetries (*Dorion et al., 2000*). To date, no studies have investigated the direct relationships between the interindividual variability in hemispheric specialization for language and anatomical hemispheric asymmetries or *corpus callosum* volume, so there is still no evidence of a direct association between anatomical and functional asymmetries.

This survey of previous findings can be summarized in the following way: leftward lateralized typical individuals can be right- or left-handers, they have leftward anatomical asymmetries both at the hemispheric level and at the regional level, they are leftward asymmetrical during language production and leftward asymmetrical for Rs_DC at rest, and they show lower intrinsic interhemispheric connectivity than individuals who are symmetrical during language production. In contrast, the type of organization in individuals who are not leftward lateralized is difficult to summarize because of the low incidence of atypical language lateralization and its heterogeneity. For instance, either a rightward asymmetry or an absence of asymmetry during language production can be observed in atypical individuals, corresponding either to a shift in the dominant hemisphere or to bilateral dominant or nondominant hemispheres, as we have previously shown using support vector machine (*Zago et al., 2017*). Moreover, to our knowledge, there is a lack of information on the lateralization of language comprehension and reading and the occurrence of dissociations in relation to typical or atypical language organization.

Another open question is the relationship between typical and atypical hemispheric specialization for language and cognitive performance that can be envisioned within different frameworks. Some authors have proposed that the decreased verbal performance observed in language developmental disorders may be related to a lack of lateralization (*Bishop, 2013*; *Tallal, 1981*; *Tallal and Schwartz, 1981*). In healthy volunteers, there is some evidence of such a relationship but with a moderate impact on performance not specific to verbal abilities (*Mellet et al., 2014b*). A larger framework would be the possible association between defects in complementary specialization and nonoptimal cognitive functioning (reviewed in *Vingerhoets, 2019*), with such an abnormal setting of complementary specialization being expected to occur in individuals with atypical hemispheric specialization for language brain organization.

To address these important questions regarding the interindividual variability in language organization in healthy individuals, we investigated 287 participants from the BIL and GIN who completed both resting-state fMRI and task-related fMRI during sentence-production, sentence-listening and sentence-reading tasks. These participants were also mapped for their anatomical hemispheric asymmetries and completed a battery of seven verbal and four visuospatial tests (*Mazoyer et al., 2016*).

## Results

### Descriptive statistics of the groups identified by hierarchical clustering

The agglomerative hierarchical procedure resulted in the identification of 3 clusters; three clusters were found to be optimal by 14 R statistical indices (from over 30 that were used to assess the quality of the classification). Hereafter, we will refer to these clusters as groups varying in their 'language organization'. These three clusters were labeled according to their task-induced mean asymmetries: a first cluster including 125 participants with strong leftward asymmetries in the three language tasks was named strong typical (TYP_STRONG; see *Table 1* and *Figure 1*), a second cluster of 132 participants exhibiting moderate leftward asymmetry was labeled mild typical (TYP_MILD), while the third cluster included the remaining 30 participants showing rightward mean asymmetry in the three tasks was labeled atypical (ATYP). Whenever needed, the TYP_STRONG and TYP_MILD groups were pooled and referred to as the TYP group.

### Task performance

Response time in each of the three tasks did not depend on 'language organization' (*Table 2*), when age, handedness and sex were taken into account (all p>0.49). The mean number of words generated per sentence was 12.4 (SD = 2.0), was also independent of 'language organization' classification, when age, handedness and sex were taken into account (p=0.97). Note that the average number of recalled sentences was 9.42 (SD = 0.96) for a maximum of 10.

### Demography and handedness

A significant difference was observed in the proportion of left-handers among the 3 'language organization' groups (p=0.0007) due to a larger proportion of left-handers in the ATYP (83.3%) than in either the TYP_MILD (49.3%) or TYP_STRONG (46.4%) groups. The differences in the proportion of left-handers were significant between the ATYP and TYP_MILD groups (p=0.0007) and ATYP and

**Table 1.** Characteristics of the three groups after hierarchical clustering on the variables that served at the classification and also on absolute values of task-induced asymmetries.

SENT_CORE network asymmetry (left minus right) was calculated as the volumetric mean of the 18 hROIs in each contrast while hub asymmetry was calculated as the volumetric mean of the 3 hROIs classified as hubs in 145 right-handers (RH)(inferior frontal gyrus: F3t, and two regions of the superior temporal sulcus: STS3 and STS4). mIHHC corresponds to the averaged resting-state Inter Hemispheric Homotopic Correlation across the 18hROIs composing SENT_CORE (Rs_mIHHC). Resting-state Degree Connectivity (Rs_DC) was calculated in the SENT_CORE network in each hemisphere. Mean Rs_DC corresponds to the mean of the left and right SENT_CORE Rs_DC. The standard deviations are between brackets.

| | TYP_STRONG N = 125 | TYP_MILD N = 132 | ATYP N = 30 |
|---|---|---|---|
| **Task-induced variables** | | | |
| SENT_CORE asymmetry | | | |
| PROD$_{SENT-WORD}$ | 0.557 (0.17) | 0.296 (0.12) | −0.114 (0.19) |
| LISN$_{SENT-WORD}$ | 0.299 (0.13) | 0.167 (0.09) | −0.155 (0.17) |
| READ$_{SENT-WORD}$ | 0.351 (0.18) | 0.217 (0.14) | −0.177 (0.15) |
| SENT_HUBS asymmetry | | | |
| PROD$_{SENT-WORD}$ | 0.80 (0.23) | 0.391 (0.18) | −0.119 (0.30) |
| LISN$_{SENT-WORD}$ | 0.42 (0.19) | 0.210 (0.15) | −0.291 (0.28) |
| READ$_{SENT-WORD}$ | 0.51 (0.29) | 0.287 (0.23) | −0.358 (0.30) |
| SENT_CORE absolute asymmetry | | | |
| PROD$_{SENT-WORD}$ | 0.557 (0.17) | 0.295 (0.12) | 0.190 (0.12) |
| LISN$_{SENT-WORD}$ | 0.300 (0.12) | 0.168 (0.08) | 0.169 (0.12) |
| READ$_{SENT-WORD}$ | 0.351 (0.18) | 0.221 (0.13) | 0.189 (0.13) |
| SENT_HUBS absolute asymmetry | | | |
| PROD$_{SENT-WORD}$ | 0.803 (0.23) | 0.393 (0.17) | 0.250 (0.20) |
| LISN$_{SENT-WORD}$ | 0.430 (0.19) | 0.214 (0.14) | 0.318 (0.25) |
| READ$_{SENT-WORD}$ | 0.515 (0.30) | 0.307 (0.20) | 0.370 (0.28) |
| Resting-state variables | | | |
| Rs_mIHHC | 0.571 (0.07) | 0.578 (0.07) | 0.610 (0.06) |
| mean Rs_DC | 8.670 (1.42) | 7.949 (1.24) | 9.460 (1.60) |
| Rs_DC asymmetry | 0.500 (0.77) | 0.478 (0.64) | −0167 (0.78) |

The online version of this article includes the following source data for Table 1:

Source data 1. Data Source for *Table 1*.

Source data 2. Detailed information concerning the *Table 1* data Source file.

TYP_STRONG groups (p=0.0001), while no difference was observed between the TYP_MILD and TYP_STRONG groups (p=0.48).

The proportion of women differed among the three groups (p=0.006, chi-square test); the proportion was significantly higher in the TYP_MILD group (58%) than in the TYP_STRONG group (38%; p=0.0013, t-test) but was not different between the ATYP group (50%) and either the TYP_MILD (p=0.41) or TYP_STRONG (p=0.25) groups.

Note that there were no significant differences in age or cultural levels between the three groups (p>0.29 in both cases).

## Profile of task-induced lateralization according to 'language organization'

A significant 'task' by 'language organization' interaction on the absolute values of task-induced asymmetry was found for both the SENT_CORE and the SENT_HUBS set of regions of interest

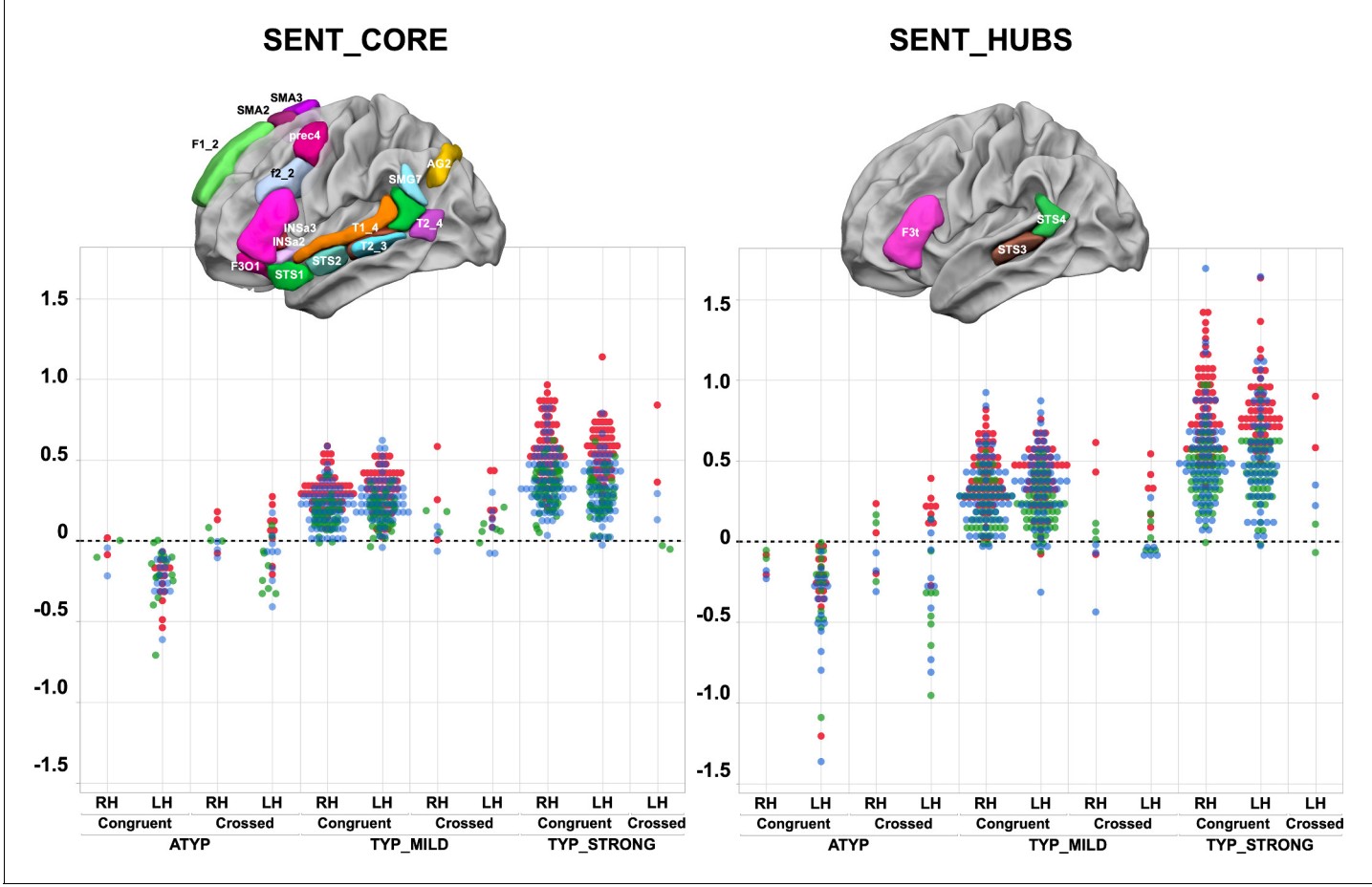

**Figure 1.** Scatterplots of individual asymmetry values in each task measured as the mean of SENT_CORE and as the mean of the three hubs (SENT_HUBS) for the three groups clustered by hierarchical clustering and stratified according to their status as CONGRUENT or CROSSED and their handedness (right-handers: RH, left-handers: LH) . The first row depicts the location of the 18 hROIs constituting the SENT_CORE network (left) and the 3 hROIs constituting SENT_HUBS (right). Atypicals (ATYP), typicals with moderate asymmetries (TYP_MILD) and typicals with strong asymmetries (TYP_STRONG) correspond to the three groups resulting from multitask and multimodal hierarchical agglomerative clustering. (PROD$_{SENT-WORD}$: red, LISN$_{SENT-WORD}$: green and READ$_{SENT-WORD}$: blue).

The online version of this article includes the following source data for figure 1:

**Source data 1.** Data source for *Figure 1*.
**Source data 2.** Detailed information concerning the *Figure 1* data source file.

(ROIs) (MANOVA analyses, $p < 10^{-4}$ for both cases; see *Figure 1* and *Table 1*). Indeed, in contrast to the two other groups, the ATYP group did not show any difference in asymmetry across the language tasks in either SENT_CORE (all $p > 0.99$) or SENT_HUBS ($p > 0.83$). In contrast, there were significant differences between tasks for both the TYP_STRONG or TYP_MILD groups in both SENT_CORE and SENT_HUBS, with a stronger asymmetry during the PROD$_{SENT-WORD}$ than during the READ$_{SENT-WORD}$ task (all $p < 0.001$), with the asymmetry during the latter being larger than that during the LISN$_{SENT-WORD}$ task (all $p < 0.017$). Note that the 'task' main effect was significant for both SENT_CORE and SENT_HUBS ($p < 10^{-4}$).

There was a significant 'language organization' by 'handedness' interaction on SENT_CORE ($p = 10^{-4}$), although the interaction did not reach significance on SENT_HUBS ($p = 0.12$). In SENT_CORE, this interaction was because the right-handed individuals in the TYP_STRONG group had higher asymmetry strength than the left-handers ($p = 0.03$), while the opposite pattern was observed in the ATYP group: left-handers had stronger asymmetry strength than right-handers (uncorrected $p = 0.0075$). Note that there were no differences between right-handers and left-handers in the

**Table 2.** Measures related to task execution in the three groups varying in hemispheric lateralization.
Mean (SD) of response times and self-reports of task difficulty rated on a 1 to 5 scale are shown for each fMRI run (Sentence production: PROD, sentence listening: LISN, sentence reading: READ). In addition, sample mean (SD) of the average number of words per sentence recalled during the debriefing of the PROD run is shown.

| | TYP_STRONG N = 125 | TYP_MILD N = 132 | ATYP N = 30 |
|---|---|---|---|
| Task difficulty | | | |
| LISN | 1.12 (0.40) | 1.14 (0.45) | 1.25 (0.80) |
| READ | 1.17 (0.40) | 1.23 (0.55) | 1.20 (0.43) |
| PROD | 2.74 (1.07) | 2.69 (1.06) | 2.88 (1.10) |
| Response time (ms) | | | |
| LISN | 388.8 (134) | 386.3 (126) | 396.9 (96) |
| READ | 3733.3 (579) | 3731.8 (560) | 3755.8 (552) |
| PROD | 5600.0 (850) | 5631.2 (968) | 5645.5 (1095) |
| Number of words per sentence | | | |
| PROD | 12.36 (2.03) | 12.37 (1.87) | 12.36 (2.55) |

The online version of this article includes the following source data for Table 2:

Source data 1. Data Source for *Table 2*.

Source data 2. Detailed information concerning the *Table 2* data Source file.

TYP_MILD (p=0.64) and ATYP (p=0.08) groups. A similar pattern, although not reaching the significance threshold, was found in SENT_HUBS.

There was no main effect of 'handedness' on the absolute values of asymmetries with SENT_HUBS (p=0.94), but there was a significant effect with SENT_CORE (p=0.0023). Finally, there was no significant 'language organization' by 'handedness' by 'task' triple interaction (SENT_CORE: p=0.29; SENT_HUBS: p=0.57).

## Intrinsic connectivity

In contrast to previous findings, there was no 'handedness' main effect or 'handedness' by 'language organization' interaction on any of the SENT_CORE intrahemispheric and interhemispheric intrinsic connectivity variables (p>0.52).

A significant main effect of 'language organization' was observed on the mean resting-state interhemispheric homotopic correlation (Rs_mIHHC, p=0.0077) due to significantly lower Rs_mIHHC in the TYP_STRONG group than in the ATYP group (p=0.01, see *Table 1*), while there were no significant differences between the TYP_STRONG and TYP_MILD groups (p=0.12) or between the ATYP and TYP_MILD groups (p=0.20).

A significant main effect of 'language organization' was observed on the average of the left and right intrahemispheric degree centrality (Rs_DC, $p<10^{-4}$): the ATYP group showed a significantly higher average Rs_DC than either the TYP_MILD ($p<10^{-4}$) or TYP_STRONG (p=0.013) groups, while the TYP_STRONG group had a significantly higher average Rs_DC than the TYP_MILD group ($p<10^{-4}$).

A 'language organization' by 'side' interaction was also found to be significant on Rs_DC ($p<10^{-4}$): the ATYP group showed no leftward asymmetry (asymmetry not significantly different from 0: p=0.80), in contrast to both the TYP_STRONG and TYP_MILD groups (significant leftward asymmetry: both $p<10^{-4}$), leading to significant differences between the ATYP group and the two other groups (both $p<10^{-4}$), while Rs_DC leftward asymmetry was not different between the TYP_MILD and TYP_STRONG groups (p=0.96).

Inspection of the right and left Rs_DC values in the three groups showed two different patterns depending on the considered hemisphere (*Figure 2* and *Table 1*). In the left hemisphere, the TYP_MILD group had a significantly lower Rs_DC than the TYP_STRONG group (p=0.0018) but was not different from the ATYP group (p=0.063), and the TYP_STRONG group was not different from the ATYP group (p=1). In the right hemisphere (*Figure 2*), the ATYP group had very strong Rs_DC

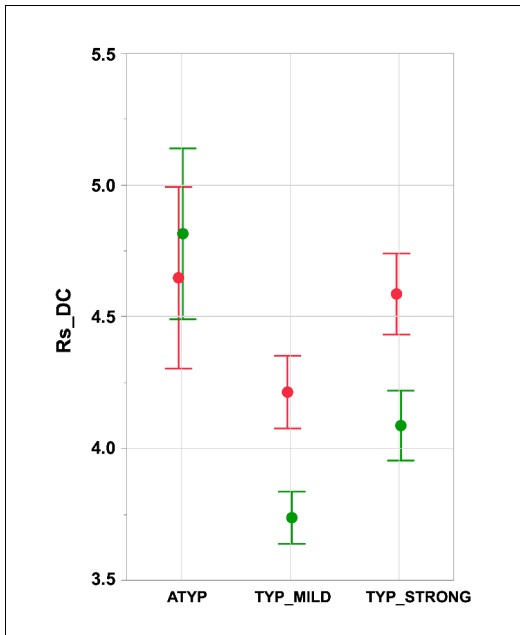

**Figure 2.** Intrahemispheric mean intrinsic connectivity strength of the SENT_CORE network in the three groups differing in language organization. Right (green) and left (red) values of the mean resting-state degree connectivity (Rs_DC) of SENT_CORE in the three groups. Significant leftward DC asymmetry is only present in TYP groups (Tukey's HSD test p<$10^{-4}$, N TYP_MILD = 132, N TYP_STRONG = 125) and right Rs_DC is higher in the ATYP group (N = 30) than in the TYP_STRONG and TYP_MILD groups (p<$10^{-4}$, Tukey's HSD test). Error bars correspond to the 95% confidence intervals.

The online version of this article includes the following source data for figure 2:

**Source data 1.** Data source for *Figure 2*.
**Source data 2.** Detailed information concerning the *Figure 2* data source file.

values, which were larger than those in both the TYP_MILD (p<$10^{-4}$) and TYP_STRONG (p<$10^{-4}$) groups, whereas the TYP_STRONG group showed significantly larger Rs_DC values than the TYP_MILD group (p=0.0044).

## Dissociations in asymmetry direction across tasks

### Descriptive statistics

Twenty-three individuals exhibited dissociation in their asymmetries induced by the three language tasks. These 23 individuals will be referred to as 'CROSSED' and the others as 'CONGRUENT'. The occurrence of CROSSED individuals within each lateralization group was higher in the ATYP group (N = 12, 40%) than in either the TYP_MILD (N = 9, 6.82%, p<$10^{-4}$) or TYP_STRONG (N = 2, 1.6% p<$10^{-4}$) groups, while the difference in the occurrence of dissociation between the TYP_MILD and TYP_STRONG groups failed to reach significance (p=0.057).

Seventeen of the 23 (74%) CROSSED participants were left-handed, a proportion significantly larger than that in the rest of the sample (p=0.02). Meanwhile, the gender ratio was not different from the rest of the sample (10 women, 43%; p=0.60).

Dissociations in the CROSSED_ATYP individuals mostly corresponded to leftward asymmetry during PROD$_{SENT-WORD}$ together with rightward asymmetry during READ$_{SENT-WORD}$ and LISN$_{SENT-WORD}$, and this pattern held for both SENT_CORE and SENT_HUBS (*Figure 3*). Only 3 of the 12 CROSSED_ATYP individuals showed the reverse pattern of rightward asymmetry during PROD$_{SENT-WORD}$ together with leftward asymmetry during READ$_{SENT-WORD}$ and/or LISN$_{SENT-WORD}$ (see *Figure 3*).

The picture was very different for dissociations in TYP_MILD individuals who were characterized by small rightward asymmetries mainly observed with READ$_{SENT-WORD}$ (only one participant had a strong negative asymmetry with READ$_{SENT-WORD}$ in SENT_HUBS). Finally, the two dissociations observed in the TYP_STRONG group were very weak negative asymmetries during LISN$_{SENT-WORD}$ regardless of the considered set of ROIs (see *Figure 3*).

There was a main effect of 'dissociation' on the task-induced strength of asymmetry restricting the analysis to the ATYP and TYP_MILD groups, where DISSOCIATED had lower asymmetry strength than CONGRUENT in both SENT_CORE and SENT_HUBS (both p<0.013), without any interaction with"task' or"language organization' (all p>0.20).

### Dissociation and resting-state organization

Considering the TYP individuals as a single group because they did not show any difference in Rs_DC (i.e. the TYP_MILD and TYP_STRONG groups were merged), there was a significant 'language organization' by 'dissociation' interaction on the mean Rs_DC value (p=0.049) due, in

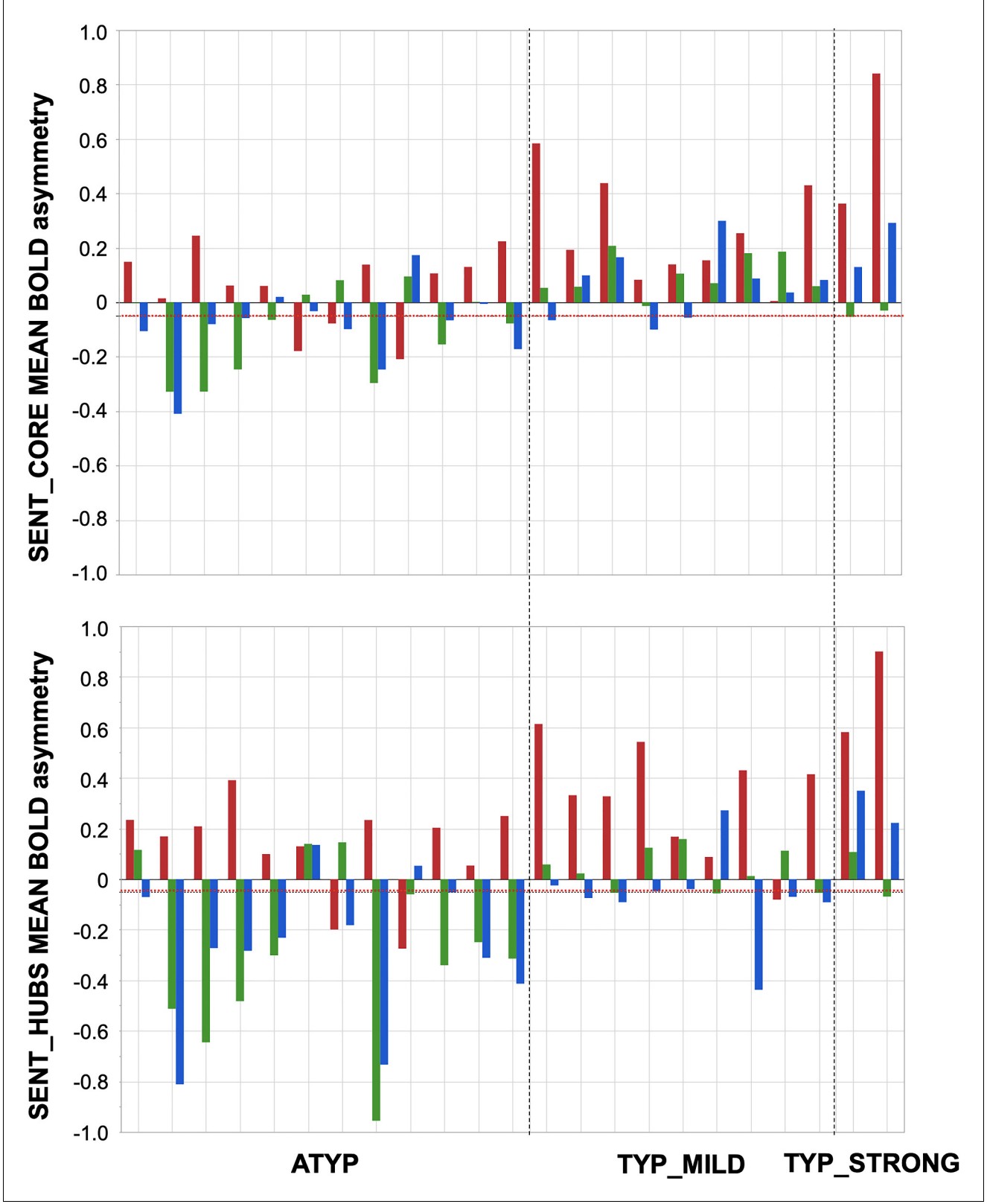

**Figure 3.** Participants showing dissociations between their three task-related functional asymmetries in each of the three groups classified by language organization. Individual values of left minus right blood oxygen level-dependent (BOLD) asymmetries measured during PROD$_{SENT-WORD}$ (red), LISN$_{SENT-WORD}$ (green) and READ$_{SENT-WORD}$ (blue) in SENT_CORE regions (top) and in SENT_HUBS (bottom). The red dotted line corresponds to the arbitrary threshold of 0.05 in asymmetry strength that was applied to define a rightward asymmetry.

*Figure 3 continued on next page*

*Figure 3 continued*

The online version of this article includes the following source data for figure 3:

**Source data 1.** Data source for *Figure 3*.
**Source data 2.** Detailed information concerning the *Figure 3* data source file.

particular, to significantly higher mean Rs_DC in the CROSSED_ATYP individuals than in the CROSSED and CONGRUENT TYP individuals (p<0.0027, for all post hoc tests corrected for multiple comparisons). The CONGRUENT_ATYP individuals did not differ from the CROSSED or CONGRUENT TYP individuals (all p>0.15), and there was no difference between the CROSSED_TYP and CONGRUENT_TYP individuals (p=0.92). Note that there was no 'dissociation' main effect (all p>0.18) and no 'language organization' by 'dissociation' by 'side' interaction (p interaction = 0.60).

In contrast, there was a significant 'language organization' by 'dissociation' interaction on Rs_mIHHC (p=0.02, see *Figure 4*) due, in particular, to significantly higher Rs_mIHHC in the CROSSED_ATYP individuals than in the CROSSED_TYP individuals (merging TYP_MILD and TYP_STRONG) that were not different in Rs_mIHHC whether CROSSED or CONGRUENT (p<0.016 for all, post hoc tests corrected for multiple comparisons). The CONGRUENT_ATYP individuals did not differ from the CROSSED or CONGRUENT TYP individuals (both p>0.43), nor did the CROSSED_TYP and CONGRUENT_TYP individuals differ (p=0.91).

## Hemispheric anatomical asymmetries and corpus callosum volume

The tissue compartment values for the four groups (TYP or ATYP by CROSSED or CONGRUENT) are provided in *Table 3*. Repeated measures MANCOVA of the GMasym and WMasym residuals (after adjusting these variables for sex, handedness, age, and total intracranial volume) showed a significant main effect of 'language organization' (p=0.02). Post hoc t-tests showed that both GMasym and WMasym were smaller in the ATYP group than in the TYP group (p=0.03). There was no effect of 'dissociation' (p=0.99) and no significant 'language organization' by 'dissociation' interaction (p=0.36). There was no interaction between 'tissue compartment' (gray or white matter) and 'language organization' (p=0.92) or between 'tissue compartment' and 'dissociation' (p=0.26), and there was no 'tissue compartment' by 'language organization' by 'dissociation' triple interaction (p=0.15).

ANOVA of the CCvol residuals (after adjustment for the same covariates as for GMasym and WMasym) showed a significant 'language organization' by 'dissociation' interaction (p=0.049). Post hoc analyses showed that the CROSSED_ATYP individuals had a larger CCvol volume than the CROSSED_TYP individuals (uncorrected post hoc t-test: p=0.037, HSD correction: p=0.16; see *Table 1*).

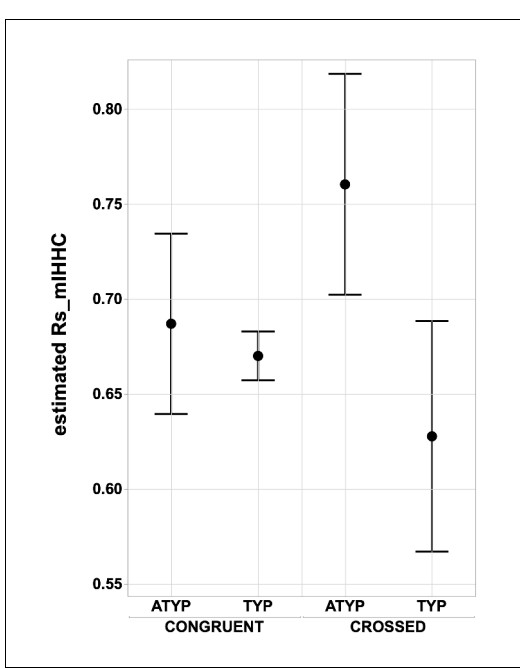

**Figure 4.** Interhemispheric intrinsic connectivity strength across homotopic regions (Rs_mIHHC) in SENT_CORE in the CONGRUENT and CROSSED TYP and ATYP groups. The estimated mean interhemispheric homotopic correlation expressed as the Fisher z-transformation of Rs_mIHHC is higher in the CROSSED atypicals group (N = 12) than in the TYP group (merging TYP_MILD and TYP_STRONG, N CROSSED = 11, N CONGRUENT = 246), regarding of whether they are CONGRUENT or CROSSED (both p<0.016, Tukey's HSD test).

The online version of this article includes the following source data for figure 4:

**Source data 1.** Data source for *Figure 4*.
**Source data 2.** Detailed information concerning the *Figure 4* data source file.

## Cognitive abilities

### Results of principal component analysis (PCA) of the 11 scores on the verbal and visuospatial tests

The average scores for the 11 completed tests are presented for each group in *Table 4*. PCA applied to the residuals of the scores (after adjustment for age, sex, cultural level and handedness) uncovered four components that explained 49% of the total variance. The first component, which we will refer to as spatial (SPA), aggregated residuals of the mental rotation test, the Corsi block test, the maze test, and the Raven matrices (loadings: 0.62, 0.39, 0.60, and 0.68, respectively). The second component, labeled phonological (PHONO), mainly included the pseudoword and rhyming test residuals (loadings: 0.48 and 0.72, respectively) and, marginally, the vocabulary test (loading: 0.36). The third component was mostly an auditory verbal memory component (MEM), including the auditory verbal word and pseudoword learning test residuals (loadings: 0.77 and 0.49, respectively). The fourth component was a verbal component (VERB) including all the verbal test residuals except those of the two learning tests, with the strongest loading being for the verb fluency test (0.64) and comparable loadings for each of the others (reading span test: 0.34, listening span test: 0.31, and vocabulary test: 0.31).

### Cognitive skills and language organization

Repeated measures MANOVA of the four PCA components (SPA, MEM, PHONO, and VERB) revealed a significant 'language organization' by 'cognitive component' interaction (p=0.0003; *Figure 5*), while the 'language organization' main effect was not significant (p=0.21).

Post hoc analyses showed that the 'language organization' by 'cognitive component' interaction was due to the difference in variation in SPA and MEM. The SPA scores were significantly higher in the TYP_STRONG group than in the two other groups, but the scores were not significantly different between the latter (TYP_-STRONG: 0.29 ± 0.15; TYP_MILD: −0.19 ± 0.14; ATYP: −0.41 ± 0.29; uncorrected p<0.0063; TYP_MILD vs. ATYP, p=0.39). Meanwhile, the MEM scores were significantly lower in the ATYP group than in the two other groups (ATYP: −0.57 ± 0.23; TYP_MILD: 0.19 ± 0.11; TYP_-STRONG: 0.05 ± 0.11; p<0.043). In addition, there was no effect of the 'language organization' on the other two verbal components, namely, VERB and PHONO (p>0.18 and p>0.13, respectively).

Finally, there was no relationship between dissociations and cognitive performance in either the ATYP or TYP_MILD individuals (p=0.17).

### Comparison of different classifications for language lateralization

We compared the outcome of the present multi-task multimodal hierarchical classification applied to the sample of 287 participants to those previously obtained with two different classifications based on the PROD$_{SENT-WORD}$ contrast only; these classifications included (1) a Gaussian mixture modeling of the HFLI observed for this contrast (*Mazoyer et al., 2014*) and (2) support

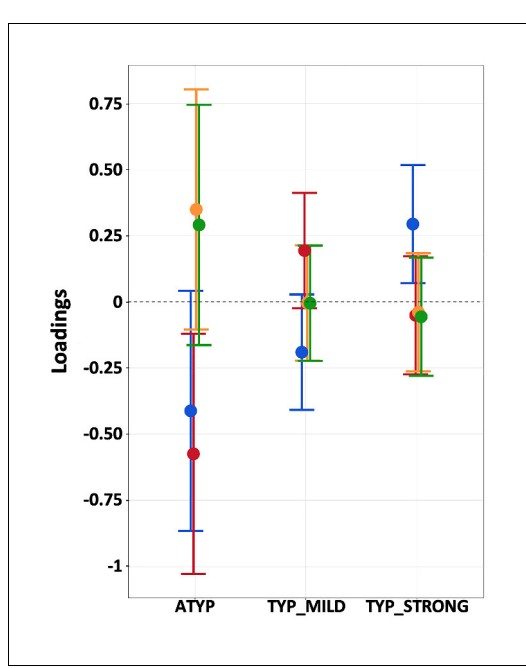

**Figure 5.** Estimated loadings of the four main principal components of cognitive abilities in the three groups having different language lateralization. The color code for the components is as follows: SPA: blue, MEM: red, PHONO: light orange, and VERB: green. Error bars represent 95% confidence intervals.

The online version of this article includes the following source data for figure 5:

**Source data 1.** Data source for *Figure 5*.
**Source data 2.** Detailed information concerning the *Figure 5* data source file.

vector machine classification of each hemisphere dominance based on the pattern of its voxels in the PROD$_{SENT-WORD}$ contrast maps (*Zago et al., 2017*). The outcomes of the multitask multimodal hierarchical classification, Gaussian mixture modeling, and support vector machine classifications applied to the same sample of 287 participants are presented in *Figure 6*.

## Multitask multimodal hierarchical classification vs. Gaussian mixture modeling

There was a high concordance of classification of typicals with the two methods (98% of Gaussian mixture modeling typicals were classified as TYP_MILD or TYP_STRONG).

Gaussian mixture modeling identified 10 rightward lateralized LH (strong atypicals; GMM-SA in *Figure 6*), among whom nine completed the resting-state acquisition and were thus included in the present study. These nine individuals were all clustered in the ATYP group as defined by the multi-task multimodal hierarchical classification. We also found that 15 other individuals in the multitask multimodal hierarchical classification ATYP group actually belonged to the ambilateral group identified by Gaussian mixture modeling (GMM-AMB in *Figure 6*) according to their weak PROD$_{SENT-WORD}$ HFLI. The remaining six individuals in the multitask multimodal hierarchical classification ATYP group were classified by Gaussian mixture modeling as typicals (GMM-TYP in *Figure 6*) because of their leftward HFLI during PROD$_{SENT-WORD}$.

The multitask multimodal hierarchical classification did not individualize any cluster resembling the group of 37 ambilaterals as defined by Gaussian mixture modeling in the *Mazoyer et al., 2014* study. Rather, aside from the 15 aforementioned ambilaterals clustered in the ATYP group in the present study, the 22 other ambilaterals as defined by Gaussian mixture modeling were here classified either as TYP_MILD (N = 16) or TYP_STRONG (N = 6).

Note that whereas all Gaussian mixture modeling-SA were left-handers, 5 among the 135 right-handers (3.7%) were classified as atypical with multitask multimodal hierarchical classification: two among these were dissociated with leftward lateralization during language production in SENT_-CORE and the SENT_HUBS, leaving only three right-handers with atyical organization in the three tasks (2%). These three right-handers were not classified as strong-atypical by Gaussian mixture

**Table 3.** Gray and white matter hemispheric volumes and their left minus right asymmetry (mean and (SD), in cc) as well as midsagittal corpus callosum volume (mean and (SD), in cc), in subgroups of individuals according to their multitask multimodal hierarchical classification and the absence/presence of dissociated task-related functional asymmetries.

TYP: participants classified with multitask multimodal hierarchical classification as either TYP_STRONG or TYP_MILD, that is showing TYP left functional lateralization; ATYP: participants classified with multitask multimodal hierarchical classification as ATYPICAL, that is showing atypical right functional lateralization. CROSSED: participants with at least one dissociation of functional lateralization among the three language tasks; CONGRUENT: participants with no dissociation.

| | TYP | | ATYP | |
|---|---|---|---|---|
| | Congruent N = 246 | Crossed N = 11 | Congruent N = 18 | Crossed N = 12 |
| Gray Matter | | | | |
| Left | 327.00 (32) | 337.59 (47) | 337.42 (35) | 317.30 (16) |
| Right | 315.64 (32) | 325.72 (48) | 327.77 (33) | 306.48 (15) |
| Asymmetries | 11.35 (4.00) | 11.87 (4.50) | 9.65 (5.03) | 10.82 (3.82) |
| White Matter | | | | |
| Left | 216.67 (26) | 221.92 (36) | 222.43 (26) | 208.61 (15) |
| Right | 213.38 (25) | 217.82 (35) | 220.22 (25) | 206.38 (14) |
| Asymmetries | 3.33 (2.19) | 4.11 (1.74) | 2.22 (2.41) | 2.23 (2.03) |
| *Corpus Callosum* | 5.31 (0.85) | 5.21 (0.67) | 5.45 (0.90) | 5.48 (0.74) |

The online version of this article includes the following source data for Table 3:

Source data 1. Data Source for *Table 3*.

Source data 2. Detailed information concerning the *Table 3* data Source file.

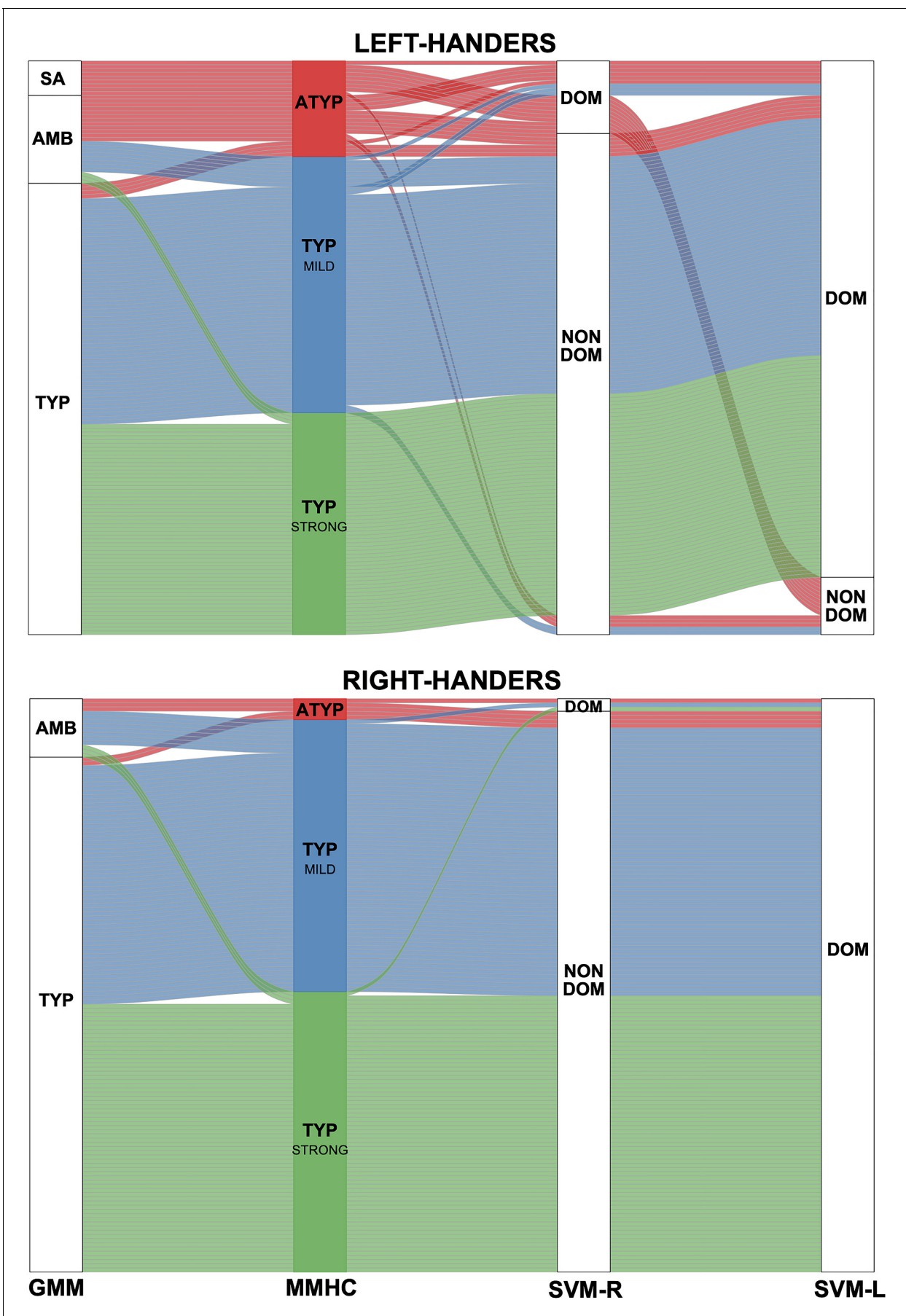

**Figure 6.** Alluvial plots comparing the present Multitask Multimodal Hierarchical Classification (MMHC) with two previous classifications only based on the functional asymmetry during production of sentences minus word-list in the same sample of participants: Gaussian Mixture Modeling (GMM) classification on Hemispheric Functional Lateralization Index (HFLI, *Mazoyer et al., 2014*) and Support Vector Machine (*Zago et al., 2017*) classification in the right (SVM-R) and left hemisphere (SVM-L). Each line corresponds to a participant with the following color code: red for multitask multimodal hierarchical classification-atypical (ATYP), blue for multitask multimodal hierarchical classification-TYP_MILD, and green for multitask multimodal hierarchical classification-TYP_STRONG. The Gaussian mixture modeling method identified each individual as either strong_atypical (SA), ambilateral (AMB), or typical (TYP). identified the voxel-based pattern of each hemisphere of an individual as either dominant (DOM) or nondominant (NON DOM).

The online version of this article includes the following source data for figure 6:

**Source data 1.** Data source for *Figure 6*.
**Source data 2.** Detailed information concerning the *Figure 6* data source file.

---

modeling but rather as ambilaterals because their HFLI for PROD$_{SENT-WORD}$, albeit negative, was above the threshold ($-50$) used for segregating strong atypicals from ambilaterals.

## Multitask multimodal hierarchical classification vs. support vector machine

Seventeen of the 30 atypical individuals as defined by multitask multimodal hierarchical classification (57%) had a right-hemisphere labeled dominant by support vector machine. Conversely, the multitask multimodal hierarchical classification atypical cluster aggregated 77% of the 22 participants labeled as having a dominant right hemisphere. One should also note that 41% (7 among 17) of these right-hemisphere dominant ATYP individuals also had a left dominant hemisphere (i.e. were codominant), whereas the ATYP cluster aggregated 77% of the 12 participants labeled as having a codominant hemisphere. Notably, the eight ambilaterals as defined by Gaussian mixture modeling left-handers classified as having a dominant right-hemisphere pattern were classified as atypical by multitask multimodal hierarchical classification (*Zago et al., 2017*).

**Table 4.** Mean (SD, in cc) of scores at the different tests of the cognitive battery in the three groups differing in their language organization as defined by a multitask multimodal hierarchical classification.

| | TYP_STRONG<br>N = 125 | TYP_MILD<br>N = 132 | ATYP<br>N = 30 |
|---|---|---|---|
| Verbal tests | | | |
| Rey: word learning | 65.98 (7.57) | 65.07 (7.51) | 64.73 (7.57) |
| Pseudo-words learning | 36.52 (10.44) | 34.64 (11.16) | 34.47 (9.68) |
| Verbal fluency | 48.19 (9.82) | 46.55 (10.01) | 47.63 (8.85) |
| Reading span test | 4.07 (1.08) | 3.91 (1.11) | 4.25 (1.11) |
| Listening span test | 4.85 (1.06) | 4.57 (1.14) | 4.63 (1.21) |
| Vocabulary | 28.39 (3.78) | 27.80 (3.74) | 28.07 (4.42) |
| Rhyming | 68.38 (4.56) | 67.17 (6.11) | 65.93 (5.62) |
| Visuo-spatial tests | | | |
| Mental Rotation Test | 11.08 (4.27) | 10.70 (4.60) | 10.17 (4.63) |
| Corsi block | 5.99 (1.06) | 5.72 (1.04) | 5.73 (0.94) |
| Maze | 6.68 (2.66) | 6.09 (2.32) | 4.42 (2.39) |
| Raven matrix | 111.78 (9.70) | 109.82 (10.47) | 106.00 (9.48) |

The online version of this article includes the following source data for Table 4:

**Source data 1.** Data Source for *Table 4*.

**Source data 2.** Detailed information concerning the *Table 4* data Source file.

## Summary of the results

In a sample of 287 healthy adults that included over 50% left-handers, a hierarchical classification based both on language task-induced asymmetries and on resting-state organization within the SENT_CORE network identified three clusters of individuals with different intra- and interhemispheric organization for sentence processing. Two clusters of similar sizes aggregated 257 (90% of the sample) leftward lateralized individuals. The 132 TYP_STRONG individuals (of which 46.4% were left-handers) were highly leftward lateralized for both task-induced asymmetry and intrahemispheric intrinsic connectivity, while showing low interhemispheric connectivity. This pattern of language organization was associated with strong leftward asymmetry of gray and white matter hemispheric volumes and with high visuospatial performance. The 125 TYP_MILD individuals (including 50.7% left-handers) differed from the TYP_STRONG individuals by their moderate leftward task-induced asymmetries, lower left hemisphere degree of connectivity and larger interhemispheric homotopic connectivity. The moderate leftward language organization in the TYP_MILD individuals was more frequent in women and was associated with a larger occurrence of dissociations than in the TYP_-STRONG individuals (7% compared to 1.6%). Visuospatial cognitive abilities were lower in the TYP_-MILD group than in the TYP_STRONG group. The third (ATYP) cluster of 30 individuals included the highest proportion of left-handers (83%). Mean asymmetry in the ATYP group was rightward lateralized during the three language tasks, with a striking lack of differences in asymmetry strengths across tasks, in contrast to the two groups of typicals. Organization at rest in the ATYP group was marked by bilateral high intrahemispheric connectivity and strong interhemispheric connectivity. Such a low hemispheric specialization pattern was associated with a high occurrence of dissociations among the functional asymmetries in the three language tasks (40%), lower leftward asymmetries of gray and white matter hemispheric volumes, and when dissociated, larger corpus callosum volumes. Finally, the ATYP cluster showed lower verbal memory abilities than the two other clusters. Comparison of the present classification to previous classifications based only on $PROD_{SENT-WORD}$ revealed the importance of the multitask approach conjointly with resting-state measures of Rs_DC in the language network to segregate the atypicals within the individuals with low $PROD_{SENT-WORD}$ hemispheric asymmetries.

## Discussion

### A multimodal multitask classification provides an enhanced definition of atypical language organization

Compared to the high consistency of the classification of individuals having typical language organization, the definition of atypicality for language lateralization based on neuroimaging investigations is complex, and the type of brain organization supporting language functions in atypical individuals is still not comprehensively understood.

All individuals having a rightward hemispheric lateralization of language production as measured with Gaussian mixture modeling were classified into the ATYP group in the present study, suggesting that having a rightward lateralization for production is a clear criterion of atypicality, as already validated by Wada studies (*Dym et al., 2011*). However, the present classification did not individualize a cluster resembling the group of 37 ambilaterals identified in *Mazoyer et al., 2014*, which indicated that not being clearly lateralized by production was not sufficient to ascertain atypicality. The difficulty in asserting language dominance in individuals with little fMRI lateralization during production is consistent with Bauer et al.'s meta-analysis showing that fMRI is more accurate in assessing language dominance in cases of strong leftward asymmetry (*Bauer et al., 2014*). However, the Bauer study involved patients suffering from epilepsy and thus likely to have language network reorganization.

To identify the discriminative variables that split the 37 ambilaterals into the 3 'language organization' groups, we conducted an additional analysis entering the nine variables we used for the multitask multimodal hierarchical classification as repeated measures, and we found in the 37 individuals classified ambilaterals in *Mazoyer et al., 2014* a very significant 'language organization' main effect and interaction with the repeated measures (both p<0.0001). Post hoc analyses revealed that among these 37 ambilaterals, the 12 individuals classified as ATYP had significantly lower task-induced asymmetry in SENT_HUBS and SENT_CORE (all p<0.001) and a significantly lower Rs_DC asymmetry

(p=0.002) than those classified as TYP_MILD or TYP_STRONG. In contrast, there was no difference in averaged Rs_DC or Rs_mIHHC values. These findings thus confirmed that, in order to comprehensively describe the dominance for language in individuals having low HFLI during language production, it is useful to apply a multitask battery as has been proposed by some authors (*Baciu et al., 2005*; *Niskanen et al., 2012*), which particularly allows the detection of individuals with dissociations as demonstrated by *Baciu et al., 2003*. Importantly, the present study also demonstrated that resting-state connectivity variables, measured at the language network level, particularly Rs_DC asymmetry, in association with task-induced asymmetry, are of interest for the identification of atypical individuals.

In left-handers, a weak functional asymmetry during language production makes atypical organization with rightward asymmetry for other language components highly probable (80%), whereas the same weak functional asymmetry in right-handed individuals is associated with a TYP leftward lateralization in most cases (86%). Such observations are consistent with the classification by support vector machine showing that all ambilateral right-handers had a left hemisphere with a dominant pattern (*Zago et al., 2017*), whereas the eight left-handed ambilaterals as classified with Gaussian mixture modeling and who had a dominant right-hemisphere pattern with support vector machine, were classified into the ATYP group by the present method.

The fact that 80% of the ATYP individuals were left-handed is consistent with previous research showing that reverse lateralization is mainly seen in left-handers, whether in adults (*Króliczak et al., 2016*; *Somers et al., 2015a*) or in children (*Szaflarski et al., 2012*). Here, a right shift in hemisphere dominance for language was found in 12% of left-handers (when taking into account the different language components, as done in the present study), compared to only 6% when considering their HFLI for production only (*Mazoyer et al., 2014*). The difference between these two proportions provides an estimate of the decreased sensitivity when detection of atypicals among left-handersLH is performed using a production task only rather than a multitask multimodal approach as we implemented in the present study. The present multimodal classification identified five right-handed ATYP individuals (3.6%, compared to 16.7% of left-handers), a phenomenon as rare as the published case reports of crossed aphasia in right-handers (*Alexander and Annett, 1996*; *Hindson et al., 1984*), raising the question of whether this is a pathological state rather than part of interindividual variability of language organization (*Coppens et al., 2002*). Among these five right-handed ATYP individuals, three had been previously classified as ambilaterals and two as typicals by Gaussian mixture modeling of the PROD$_{SENT-WORD}$ HFLI, with the latter two individuals having negative asymmetries during LISN$_{SENT-WORD}$ and READ$_{SENT-WORD}$. It is noticeable that these five right-handers had lower task-induced asymmetry strength than the 25 left-handed ATYP individuals, independent of the task, leaving open the question of whether right- and left-handed ATYP individuals are actually comparable.

Finally, the present classification sheds some light on the brain organization for language in individuals as defined by the support vector machine approach (*Zago et al., 2017*). Actually, the ATYP cluster aggregated 77% of the 22 participants labeled as having a right dominant hemisphere by support vector machine. This is very consistent with the high Rs_DC found for the right SENT_CORE network of the ATYP individuals. One should also note that 41% (7 among 17) of these right-hemisphere dominant ATYP individuals also had a dominant left hemisphere (i.e. were codominant), whereas the ATYP cluster aggregated 77% of the 12 participants labeled as having a codominant hemisphere. This strong association between atypicality and codominance is also consistent with the finding that ATYP individuals were characterized by high bilateral connectivity of their SENT_CORE network, which is likely to reduce the bias toward the dominance of a given hemisphere and attest to a more bilateral organization for language.

## Organization of intrinsic connectivity in atypical individuals: although they show rightward task-induced asymmetries, their left hemisphere is also wired for language

In a previous study, we noted that the 10 left-handers with strong rightward HFLI exhibited a pattern of regional asymmetries that was the reverse of the pattern observed in typical individuals (*Tzourio-Mazoyer et al., 2016*), a result in line with cortical stimulation findings suggesting that individuals shifting their dominant hemisphere actually have a reverse regional organization (*Chang et al., 2011*; *Drane et al., 2012*). The present study results, although consistent with this view in terms of

task-induced asymmetries, demonstrated that, by contrast, the SENT_CORE network intrinsic connectivity properties of ATYP individuals did not mirror those of individuals with leftward task-induced asymmetries. Although the mean of the group was strongly rightward asymmetrical in the three tasks, the ATYP individuals showed high and symmetrical Rs_DC values, meaning that the SENT_CORE network was highly connected in both hemispheres, and it is remarkable that their left hemisphere Rs_DC value was not different from that of the TYP_STRONG individuals, whereas their right hemisphere Rs_DC value was higher than that of the TYP_STRONG individuals (*Figure 7*). The ATYP individuals thus had a significantly larger mean Rs_DC value of SENT_CORE in both hemispheres, making them highly connected individuals and suggesting that their left hemisphere could be organized in a way similar to that of the TYP individuals, that is as a potentially dominant hemisphere for language. In addition, the ATYP individuals showed the highest interhemispheric connectivity across SENT_CORE homotopic areas, constituting a highly efficient network for sentence processing that straddled the two hemispheres. The fact that even in individuals shifting their task-induced lateralization to the right, the left hemisphere is wired for high-order language processing leads to the hypothesis that the left hemisphere is the language hemisphere by default.

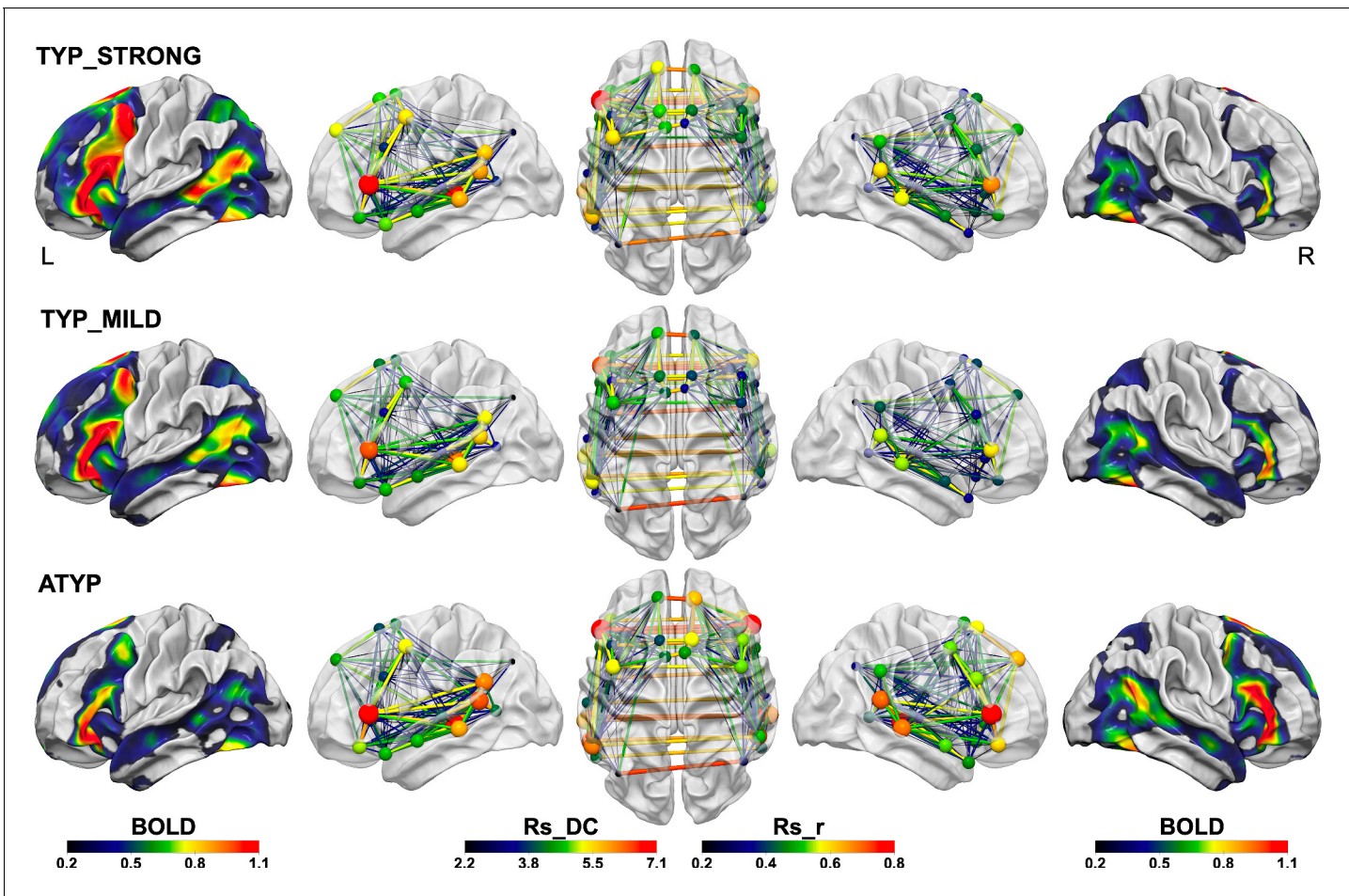

**Figure 7.** Summary figure illustrating the different SENT_CORE intra- and inter-hemispheric organizations observed in the three groups identified by hierarchical clustering. The left column shows the group mean activation maps during PROD_SENT-WORD (BOLD activation amplitude is given by color scale) of the left hemisphere and the right column the mean activation map of the right hemisphere superimposed on the white matter surface rendering of the BIL and GIN template obtained with the Surf Ice software (https://www.nitrc.org/projects/surfice/). The second, third and fourth columns show the left lateral, superior and right lateral views of the SENT_CORE intrinsic connectivity network, each region of the network being represented by a sphere located at the mass center of its MNI coordinates. For each SENT_CORE region, a colored sphere indicate the group average region degree centrality of intrinsic connectivity (the Rs_DC value is given by color scale, and sphere size is proportional to value), whereas a colored line indicates the strength of the Pearson intrinsic correlation coefficient between two SENT_CORE regions (the Rs_r value is given by color scale, and line thickness is proportional to value).

A trace of how ATYP individuals overcome the left hemisphere default-mode organization for language can be found in the loss of congruence in the sentence network at rest and during sentence processing. In right-handers, we observed a positive correlation across individuals between asymmetries of activations and Rs_DC (*Labache et al., 2019*), while the ATYP group showed an absence of mean Rs_DC asymmetry but mean rightward task-induced symmetries. Notably, both the CROSSED and CONGRUENT ATYP individuals had a left hemisphere Rs_DC as strong as that in the TYP_STRONG individuals, meaning that their left SENT_CORE network connectivity was not different from that of strong leftward lateralized individuals (*Figure 7*) supporting the hypothesis that ATYP left hemisphere is wired for language as it is for typical individuals. Actually, ATYP differed from typicals in their right-hemisphere organization at rest that exhibited a high strength of intrinsic connectivity, in agreement with their task-induced rightward activations (*Figure 7*).

The pattern of ATYP individual network intrinsic organization is thus a networking of both hemispheres profiled for the processing of high-order language, combined with strong anatomical and functional underpinning of interhemispheric interactions as evidenced by higher correlations across homotopic regions of SENT_CORE and larger corpus callosum in the DISSOCIATED_ATYP individuals. The ATYP group also showed a more bilateral anatomical organization with decreased leftward gray and white matter hemispheric asymmetries likely to result in more flexibility in the side hosting the different language tasks and therefore allowing dissociations. In fact, the ATYP group hosted the largest proportion of participants showing dissociations and thus relying on one or the other hemisphere as the dominant hemisphere depending on the language component, which may be related to their stronger interhemispheric connectivity. Such a hypothesis was partly confirmed by the comparison of individuals with dissociations in the three groups that demonstrated that CROSSED ATYP individuals had significantly higher interhemispheric connectivity and more variation in the strength of asymmetries when DISSOCIATED than the two other groups. These strong between-task differences in asymmetry strengths reflect an important shift in hemispheric control, which were particularly seen between PROD$_{SENT-WORD}$ and the two other tasks underpinned by the strong interhemispheric connectivity allowing for cooperation across the bilaterally located task-dependent dominant language networks.

## Two types of leftward organization for language, with an overrepresentation of women but not of left-handers in mildly lateralized typical individuals

The present segregation of leftward lateralized individuals in the two groups is consistent with the two Gaussian components of the PROD$_{SENT-WORD}$ HFLI distribution in typical individuals observed in our previous work (*Mazoyer et al., 2014*). However, these two Gaussian components showed too much overlap to allow a clear separation of the two groups of typical individuals. One original observation of the present study is thus the evidence of differences in terms of functional connectivity between two groups of typical individuals. Although leftward lateralized and showing the same gradient of asymmetry across the three tasks, the TYP_MILD individuals exhibited significant particularities in their inter- and intrahemispheric—although typical—intrinsic connectivity organization with lower asymmetries of task-induced activations but also lower Rs_DC and higher Rs_mIHHC within SENT_CORE. In other words, their decreased strength in task-induced functional asymmetries was associated with an intra- vs. interhemispheric intrinsic connectivity pattern showing less differentiation across hemispheres together with increased connection between them. Such a pattern of looser hemispheric specialization for language in the TYP_MILD group is consistent with a higher occurrence of dissociations than in the TYP_STRONG group, although those dissociations were of moderate intensity and mainly observed for the reading task.

The proportion of women was larger in the TYP_MILD cluster (58%) than in either of the two other clusters (38% in the TYP_STRONG and 50% in the ATYP clusters), as well as in the whole sample (49%), consistent with previous reports of reduced language lateralization in women (*Levy and Reid, 1978*; *McGlone and Davidson, 1973*). Interestingly, gender differences in cluster constitution in the present work were present only in the two groups of typicals but not in the ATYP group. Such a subtle association between sex and language lateralization may explain the inconsistency in the reports of a sex effect in hemispheric specialization for language (*Sommer et al., 2004*) since, in contrast to handedness, it is not associated with the occurrence of critical changes in language lateralization. Actually, the proportion of left-handers was not increased in the TYP_MILD group

(compared to the TYP_STRONG group), confirming that the relationship between handedness and language lateralization is better grounded in the large occurrence of left-handers among rightward lateralized individuals rather than by a decreased lateralization for language in left-handers (*Mazoyer et al., 2014*).

## Dissociations of lateralization across language components are of different natures in typical and atypical individuals with a particular status for the lateralization of reading

Dissociations were detected with higher sensitivity when considering the SENT_HUBS hROIs rather than the whole set of SENT_CORE area ROIs. This is the reason why we considered a participant dissociated if they had opposed asymmetry across tasks on either one or both variables.

The low incidence of dissociations that we observed in the TYP individuals and, in particular, in the TYP right-handers (4%) was consistent with the literature that reports rare cases of dissociations of production and comprehension in healthy right-handed participants (*Jansen et al., 2006*; *Tzourio-Mazoyer et al., 2004*). A point of interest was the occurrence of dissociation between the lateralization for reading and the lateralization for production and listening, which, to our knowledge, has not yet been reported. In leftward lateralized TYP individuals, dissociations were mainly observed in the TYP_MILD individuals for whom, as in the TYP_STRONG individuals, reading was on average more lateralized than listening (although less than production).

Dissociations in this TYP_MILD cluster more often involved reading (5 out of 9 in SENT_HUBS; see *Figure 3*). Such a larger occurrence of dissociations involving reading may be related to the late acquisition of this language function. Indeed, the first phase of language development is perceptual, as revealed by studies showing that the auditory system of the fetus at 30-week gestation is mature enough to detect complex sounds (*McMahon et al., 2012*) and to differentiate phonemes (*Hepper and Shahidullah, 1994*). After only a few hours of postnatal exposure, newborns respond specifically to speech (*Dehaene-Lambertz et al., 2002*). Then, because of maturation of the vocal tract, the second phase is production (*Mowrer, 1980*). From the second half of the first year of life, the child enters the babbling phase proper and begins to make choices specific to the structures of his or her mother tongue at the prosodic, phonetic and syllabic levels (*Oller, 1980*). These first steps toward articulation are an essential step that reflects the existence of a functional link between the processes of perception and the production of vocal sounds and gives the child the opportunity to receive proprioperceptive feedback (*Rodgon, 1976*).

While speech perception and production tightly codevelop very early in the establishment of language, reading is based on both the ability to hear and segment words into phonemes and then to associate these phonemes with graphemes, with the mapping of orthographic to phonological representations during reading being intrinsically cross-modal (*McNorgan et al., 2014*). In fact, reading develops in interaction with object recognition in the left fusiform gyrus (*Kassuba et al., 2011*) and rightward lateralized visuospatial and visuomotor processes such as the saccadic system supporting eye movement during reading (*Petit et al., 2009*). More particularly, during reading, eye movements are not only an oculomotor ability but also the integration of visual and language processes at the word level and at the syntactic level (*Richards et al., 2017*). In fact, reading depends on an alternation of fixations and saccades, the latter being defined as forward progressions or backward regressions. Even if forward progressions are the most common eye movements, backward regressions have been revealed to be correlated with the syntactic complexity of sentences, suggesting that these eye regressions depend on the relationships that the words making up the statement have to each other (*Lopopolo et al., 2019*). Thus, reading ability involves both visuospatial and language processes. Such a late specialization could lead to the possibility that different factors could intervene in the establishing of reading lateralization, with these factors being different from those acting during the first stages of language development.

The picture was very different for ATYP individuals, whose predominant dissociation pattern was a leftward lateralization for production and a rightward lateralization for both reading and listening (*Figure 3*, left). Considering the developmental timing of language components mentioned above, this could be an indication that ATYP lateralization for language perception and production is established early in different hemispheres. The second observation is that in the ATYP individuals, the lateralization of heteromodal areas during reading follows that of auditory sentence comprehension, demonstrating the prevalence of sensory integration over action in these individuals, which is

different from the lateralization organization in the TYP_MILD individuals. The fact that reading lateralization has different relationships with production and listening according to the sentence lateralization organization can provide new insight into the variability in the establishing of reading dominance and, potentially, a possible relationship between atypicality and dyslexia, since there is still a great debate between lateralization and reading impairments (*Wilson and Bishop, 2018*). Assessing the type of dissociations would be of great interest for shedding new light on language impairments.

The more frequent rightward lateralization during LISN than during PROD in the ATYP left-handers was consistent with the observation of Hécaen of a high occurrence of production deficits after left-hemisphere lesions in left-handers, while comprehension deficits were rare (*Hécaen et al., 1981*). Such a dissociation corresponds to that of action vs. perception as defined by *Fuster, 2009*, with sentence reading and listening being colateralized. It is remarkable that, when compared to both TYP groups, the ATYP group showed a decrease in (absolute value) asymmetry strength that was larger for production than for the other tasks, leading to an absence of a difference between the asymmetries in production, listening, and reading. Such a diminished asymmetry during production is striking because of the link existing between hand preference and language production, with both functions being on the action side and being localized in close frontal areas. One should have expected left-handers to have stronger rightward asymmetry during language production than during the other tasks in relation to their left-hand dominance. This was not the case, even when considering only the CONGRUENT_ATYP individuals. However, handedness was associated with a stronger mean rightward asymmetry in the left-handed ATYP individuals and stronger leftward asymmetry in the right-handed TYP_STRONG individuals, independent of the task, as if the hemisphere controlling the dominant hand is a slight attractor for language lateralization. This modest effect of handedness is consistent with the observation that patients who had suffered from right plexus brachial injury at birth, therefore disabled in the use of their right hand, present a shifting of their language production asymmetries toward the right hemisphere, although without a complete shift (*Auer et al., 2009*).

## Are different language organizations associated with differences in cognitive abilities?

Better visuospatial performance was present in the TYP_STRONG individuals, who had the largest between-hemisphere differences and lower interhemispheric connectivity. Such a result suggests that the better spatial abilities reported in RHright-handers in a meta-analysis (*Somers et al., 2015b*) might have been related to the fact that the TYP_STRONG group hosted the highest proportion of right-handers. The present results suggest that strong leftward lateralization of both language task-induced and resting-state connectivity asymmetries in the core language network is associated with better visuospatial performance, as if less involvement of the right hemisphere in sentence processing was facilitating visuospatial processing. Such an observation can be viewed as an argument in favor of the 'crowding effect' theory stating that an optimal split of functions across the two hemispheres facilitates cognitive functioning (review in *Vingerhoets, 2019*). Of course, further exploration of the relationships between the different aspects of visuospatial cognitive abilities and the strength of both leftward lateralization for language and rightward lateralization for visuospatial functions, as well as their interindividual variability, is needed to confirm this hypothesis.

Decreased verbal memory abilities in the ATYP group suggest that the reorganization occurring on top of the language organization by default in this group is at the cost of suboptimal cognitive functioning, while mild, although leftward, lateralization for language appears to be as efficient for language processing as strong leftward lateralization. Considering that the ATYP group included 15 of the ambilateral individuals defined in *Mazoyer et al., 2014*, the present observation is consistent with those of Mellet et al reporting lower performance in ambilaterals (*Mellet et al., 2014b*) concerning both verbal memory and visuospatial abilities.

## Conclusions

The joint investigation of language task-induced asymmetries and intrinsic connectivity strength in the sentence-processing supramodal network, showed that individuals with atypical rightward language lateralization do not rely on an organization that simply mirrors that of typical leftward

lateralized individuals but rather is associated with a loose hemispheric specialization for language. The fact that these individuals had lower leftward gross macroscopical hemispheric anatomy than typical individuals suggests that such organization was supported, at least in part, by early developmental events resulting from a different trajectory or from the occurrence of plastic changes. Support for the hypothesis of the early establishment of this atypical organization comes from the coinvestigation of the lateralization of production and comprehension with reading. In atypicals, dissociations were observed between sentence production and comprehension (whether read or listened to), two functions known to be tightly coupled and early developing. By contrast, the rare dissociations found in typicals occurred for reading, a later acquired competence. Moreover, atypical organization occurring mainly in left-handers has a cost in terms of language abilities with less efficient verbal memory. Finally, the present results argue for multitask measures of language lateralization for evaluating hemispheric specialization for language in individuals with low lateralization for language production, especially if they are left-handed.

## Materials and methods

### Participants

The study sample was part of the BIL and GIN database that has been fully described elsewhere (*Mazoyer et al., 2016*). Briefly, 287 healthy participants of the BIL and GIN (150 left-handed, 140 women, 72 left-handed women) who completed the fMRI battery, including several language tasks and a resting-state acquisition, were included in the present work. The sample mean age was 25.8 years (SD = 6.5 years). The mean educational level of the participants was 15.6 years corresponding to almost 5 years education after the French baccalaureate (SD = 2.3 years).

For each participant, we recorded self-reported handedness and manual preference (MP) strength assessed with the Edinburgh inventory (*Oldfield, 1971*). Left-handed participants had an Edinburgh score of −63.2 (SD = 39.9).

### Participants' cognitive evaluation

Participants' verbal abilities were evaluated with the following battery of seven tests: (1) a supraspan recall test of an 18-word-list (*Van der Elst et al., 2005*) for verbal memory evaluation; (2) a supraspan recall test of a list of 15 pseudo-words for verbal memory evaluation with minimal semantic associations; (3) a verb generation test for semantic verbal fluency exploration; (4) a synonym finding test for estimating vocabulary extent *Binois and Pichot, 1956*; (5) a listening span test based on spoken sentences; (6) a reading span test based on read sentences for verbal working memory assessment (*Daneman and Carpenter, 1980*; *Desmette et al., 1995*) and (7) a rhyming test on 80 visually presented pairs of pseudo-words for evaluation of graphophonemic conversion ability.

Visuospatial abilities were assessed with the following four tests: (1) The Mental Rotation Test (MRT), which estimates the ability to rotate and spatially manipulate mental images *Vandenberg and Kuse, 1978*; (2) the Corsi Block test, which evaluates visuospatial short-term memory abilities *Della Sala et al., 1999*; (3) a home-made 3D maze test for evaluating topographic orientation skills; and (4) the Raven matrix for assessing non-verbal reasoning.

### Language tasks completed during fMRI

The language fMRI paradigm has been fully described elsewhere (*Labache et al., 2019*). In short, three fMRI runs were completed by the participants, each including a sentence-level task and a word-list reference task corresponding to randomized alternation of event-related trials. Within each trial, the participant was shown for 1 s either a line drawing (taken from the 'Le Petit Nicolas' comic strip series) or a scrambled drawing, that was immediately followed by a central fixation crosshair. While fixating the cross, the participant performed either the sentence task or the word-list reference task.

During the production run (PROD), after seeing a line drawing, the participant was instructed to covertly generate a sentence beginning with a subject and a complement, followed by a verb describing the action taking place and ending with an additional complement of a place or a manner. When a scrambled drawing was displayed, the subject was asked to covertly generate the list of the months of the year.

During the listening run (LISN), whenever a Petit Nicolas line drawing was displayed, the subject was instructed to carefully listen to a sentence dealing with the line drawing and to click at the end of the sentence. When a scrambled drawing was displayed, he/she was instructed to listen to the list of the months, days of the week and/or seasons and click at the end of the list.

During the reading run (READ), like in the two other tasks, whenever a line drawing was displayed, the subject was instructed to read a sentence based on the outline drawing. When a scrambled drawing was displayed, he/she was instructed to read the list of months, days of the week and/ or seasons.

## Task execution and performance

The response times corresponding to the end of the sentence production, sentence listening and sentence reading were recorded for each participant during the fMRI session, and right after the fMRI session, the participants were asked to rate the difficulty of each of the tasks on a 5-level scale (1:easy to 5:very difficult). For the production run, each participant was asked to recall and write down, whenever possible, the sentence he/she elaborated when presented with each image, the average number of words per (recalled) sentence being then computed.

## Image acquisition and processing

### Image acquisition

Imaging was performed on a Philips Achieva 3 Tesla MRI scanner (Philips, Erlangen, The Netherlands).

The structural MRI protocol consisted of a localizer scan, a high resolution three-dimensional T1-weighted volume (sequence parameters: TR 20 ms; TE 4.6 ms; flip angle 10°; inversion time 800 ms; turbo field echo factor 65; sense factor 2; field of view $256 \times 256 \times 180$ mm$^3$; $1 \times 1 \times 1$ mm$^3$ isotropic voxel size) and a T2*-weighted multi-slice acquisition (T2*-weighted fast field echo (T2*-FFE), sequence parameters:TR = 3,500 ms; TE = 35 ms; flip angle = 90 deg; sense factor = 2; 70 axial slices; $2 \times 2 \times 2$ mm$^3$ isotropic voxel size).

Language task-related functional volumes were acquired using a T2*-weighted echo-planar imaging (EPI) sequence (TR = 2 s; TE = 35 ms; flip angle = 80°; 31 axial slices with a $240 \times 240$ mm$^2$ field of view and $3.75 \times 3.75 \times 3.75$ mm$^3$ isotropic voxel size). In the three runs, 192, 194, and 194 T2*-weighted volumes were acquired for the production, listening, and reading sentence tasks, respectively.

Resting-state functional volumes (N = 240) were acquired as a single 8 min long run using the same T2*-weighted EPI sequence. Immediately prior to scanning, the participants were instructed to 'keep their eyes closed, to relax, to refrain from moving, to stay awake and to let their thoughts come and go'.

### Processing of structural images

For each participant, (1) the T2*-FFE volume was rigidly registered to the T1-MRI; (2) the T1-MRI was segmented into three brain tissue classes (gray matter, white matter, and cerebrospinal fluid) and normalized to the BIL and GIN template including 301 volunteers from the BIL and GIN database using the SPM12 'segment' procedure (http://www.fil.ion.ucl.ac.uk/spm/) with otherwise default parameters. Whole volumes of these three compartments were extracted and brain volume calculated as their sum. In addition, hemispheric volumes (left and right) of gray and white matter were extracted to compute their asymmetries (*Table 3*).

A semi-automated in-house *corpus callosum* segmentation procedure was then applied to extract individual masks of corpus callosum obtained from 10 consecutive mid-sagittal slices of 1 mm width on individual white matter maps in the MNI stereotactic space. An additional processing step to remove the fornix, which was sometimes segmented and connected along with the corpus callosum, was added. Quality control of corpus callosum segmentation was achieved by visual inspection of all slices, and, when needed, manual corrections for minor segmentation error was applied using FSL software. Each individual *corpus callosum* mask was then applied to each participant's normalized and modulated white matter partition images to estimate individual corpus callosum volume (*Table 3*).

## Pre-processing of task-related and resting-state functional volumes

Functional data were corrected for slice timing differences. To correct for subject motion during the runs, all T2*-weighted volumes were realigned using a 6-parameter rigid-body registration. The EPI-BOLD scans were then registered rigidly to the structural T2*-FFE image. The combination of all registration matrices allowed for warping the EPI-BOLD functional scans from the subject acquisition space to the standard space ($2 \times 2 \times 2$ mm$^3$ sampling size) with a single interpolation.

Time series of BOLD signal variations in white matter and cerebrospinal fluid (individual average time series of voxels that belonged to each tissue class) as well as temporal linear trends were removed from the rs-fMRI data series using a regression analysis. Additionally, rs-fMRI data were bandpass filtered (0.01 Hz - 0.1 Hz) using a least-squares linear-phase finite impulse response filter design.

## Language task-fMRI processing

### Language task contrast maps

Statistical parametric mapping (SPM12, http://www.fil.ion.ucl.ac.uk/spm/) was used for processing the task-related fMRI data. First, a 6 mm full width at half maximum (FWHM) gaussian filter was applied to volumes acquired during each run. For each participant, differences between BOLD signal volumes corresponding to sentence and list belonging to the same run were computed, namely sentence minus word-list production (PROD$_{SENT-WORD}$), sentence minus word-list reading (READ$_{SENT-WORD}$), and sentence minus word-list listening (LISN$_{SENT-WORD}$).

### Regions of interest analysis using the SENSAAS atlas

BOLD signal variations during the three language tasks and resting-state and their asymmetries were then computed for the set of 18 pairs of homotopic frontal and temporal regions of interests (hROIs) that we previously identified in the subgroup of 144 right-handers as constituting a core network of language areas (SENT_CORE, *Figure 1* [*Labache et al., 2019*]). These 18 hROI-pairs were selected as activated and leftward asymmetrical in these same three tasks and as constituting at rest a network with strong positive correlations across the hROIS. Note that SENT_CORE areas contain the antero-posterior high-order language areas, consistent with language meta-analyses of healthy individuals (*Price, 2010*; *Price, 2012*; *Vigneau et al., 2006*), including three intrinsic connectivity hubs corresponding to the inferior frontal gyrus (F3t) and two regions of the superior temporal sulcus (STS: STS3 and STS4 *Figure 1*).

Here, for each participant, each of the three contrast maps, and each of these 18 hROIs, left and right BOLD signal variations were computed by averaging the contrast BOLD values of all voxels located within the hROI volume. Then, for each participant and each contrast map, mean left and right BOLD variations and asymmetry for the whole network were also computed as a weighted (by volume) average of the corresponding 18 hROIs values (SENT_CORE), as well as the mean of the three hubs (SENT_HUBS).

### Resting-state organization of SENT_CORE network

For each individual and each hROI composing the SENT_CORE network, we computed a degree centrality (Rs_DC) in each hemisphere. The Rs_DC in each participant and each hROI of each hemisphere was calculated as the sum of the positive correlations existing between one hROI and all the other hROIs of the SENT_CORE network. Rs_DC values were then averaged across the 18 hROIs of the same hemisphere and the resulting left and right averaged Rs_DC values were summed and divided by two so as to provide a SENT_CORE intra-hemispheric Rs_DC characterizing the strength of within hemisphere intrinsic connectivity for this network. We also computed the left minus right difference of the averaged Rs_DC values as a measure of the asymmetry in intra-hemispheric connectivity strengths for the SENT_CORE network.

Interhemispheric connectivity strength was estimated in each individual by the average across the 18 hROIs pairs constituting the SENT_CORE network of the z-transformed intrinsic correlation coefficient between homotopic ROIs (mean Inter-Hemispheric Homotopic Correlations, Rs_mIHHC).

## Statistical analysis

### Identification of groups of individuals with different brain organization for language through hierarchical clustering

In a previous work (*Mazoyer et al., 2014*), we have shown that the distribution of lateralization for sentence production, although of continuous nature, could be used to classify individuals into three discrete categories. So, we believe it was justified to try to categorize individuals taking into account not solely production but reading, listening and resting-state, as well. It is important to realize that we did not a priori decide that the number of categories for this multivariate classification would be 3 (as it was when using production only). Rather, the optimal number of clusters for this multivariate classification was obtained using a fully unsupervised methodology and a combination of 30 statistical criteria (see below).

The study sample was segregated in groups varying in their intra- and interhemispheric organization in SENT_CORE using an agglomerative hierarchical clustering procedure. The variables entered in this procedure were both functional asymmetries induced by each of the three language tasks and intra- and interhemispheric SENT_CORE intrinsic connectivity metrics.

Task-induced functional asymmetries were obtained both at the SENT_CORE level, because of the low intersubject variability that results from averaging over the whole set of 18 ROIs, and at the SENT_HUBS level (i.e. when averaging asymmetries over three hubs: F3t, STS3, and STS4) because, although more variable across individuals, this hub-averaged asymmetry involves supramodal regions having a key role in the sentence core network (see *Labache et al., 2019*). There were thus six variables for task-induced activation: the SENT_CORE and SENT_HUBS asymmetries for LISN-$_{SENT-WORD}$, PROD$_{SENT-WORD}$ and READ$_{SENT-WORD}$.

To investigate both the intrahemispheric integration in the language networks and the interhemispheric differences, we included in the hierarchical classification the sum of the two hemisphere Rs_DC values (left Rs_DC + right Rs_DC) and the Rs_DC asymmetry (left Rs_DC – right Rs_DC) calculated in SENT_CORE. To account for interhemispheric intrinsic connectivity strength, we included the mean of the interhemispheric SENT_CORE homotopic correlation (Rs_mIHHC).

These nine variables were standardized before being jointly entered into an agglomerative hierarchical classification (*Sneath and Sokal, 1973*) that used the Euclidean distance for computing the dissimilarity matrix and the Ward distance (*Ward, 1963*) to aggregate the different participants into clusters using the '*hclust*' function provided by default in R. The optimal number of clusters was determined using the R library '*NbClust*' (*Charrad et al., 2014*). This package provides 30 statistical indices for determining the optimal number of clusters and offers the best clustering scheme from the different results obtained by varying all combinations of the number of clusters for the chosen method, in this case, hierarchical clustering with Ward's distance. We selected the number of clusters that satisfied a maximum of indices and found it to be equal to 3.

Hierarchical classification was completed with R (R version 3.5.1; *R Development Core Team, 2013*), while other statistical analyses were performed using JMP15 (http://www.jmp.com, SAS Institute Inc, 2018).

### Identification of individuals with dissociations of lateralization across tasks

In a second step, we identified individuals exhibiting at least one dissociation in their lateralization among the three language tasks, which means those who exhibited SENT_CORE functional asymmetries larger than 0.05 in amplitude in the opposite direction in one task compared to the others. We also searched for individuals exhibiting a dissociation in their SENT_HUB asymmetries, which led to the definition of two categories of individuals: 1. those exhibiting a dissociation for either SENT_CORE or SENT_HUB or both, who were named 'CROSSED'; 2. those showing either leftward lateralization for all tasks for both SENT_CORE and SENT_HUBS or right lateralization for all tasks for both SENT_CORE and SENT_HUBS, who were named 'CONGRUENT'.

Pearson's chi-square tests were conducted to compare the proportion of 'dissociation' across the clusters identified by the classification.

## Characterization of the groups provided by the classification with different brain organization of the language network

### Task performance, demography and handedness

To ensure that potential differences in the asymmetries measured during the language tasks were not related to group differences in task execution time that was recorded during the task-induced fMRI session, response times were compared across 'language organization' groups (corresponding to the clusters of the hierarchical classification) taking into account sex, age, and brain volume. In addition, within each 'language organization' group, we compared the groups of 'CONGRUENT' and 'CROSSED' individuals.

The different 'language organization' groups were compared with variables known to be associated with variability in language lateralization, namely, handedness, sex, age, and brain volume. To complete these analyses, Pearson's chi-square tests were applied for discrete variables (handedness and sex) and ANOVA with Tukey's HSD post hoc tests for continuous variables (age and brain volume).

### Task-induced and resting-state organization of the SENT_CORE network in groups varying in their language network organization

We first comprehensively described the different types of organization of the sentence networks in the groups issued from the hierarchical classification.

We used two repeated measures MANOVA to examine the task-induced asymmetries within SENT_CORE and SENT_HUBS in the three language tasks searching for 'task' (three levels), 'language organization' (three levels, that is, a number of levels corresponding to the number of identified clusters) and 'handedness' main effects and their interactions.

Note that to ensure that this between-group difference was not due to a difference in the occurrences of dissociations across 'language organization' groups, the statistical analysis was completed on the absolute values of asymmetries within SENT_CORE and SENT_HUBS. We also examined in the ATYP and TYP_MILD groups the effect of 'dissociation' and its interactions with 'task' on the strength of task-induced asymmetries. The TYP_STRONG group was not considered in this analysis because there were only 2 DISSOCIATED individuals in this group.

In the same way, we examined the resting-state variables, that is the Rs_DC asymmetries, Rs_DC mean and Rs_mIHHC. For Rs_mIHHC, we performed the Fisher z-transformation to conduct the analysis.

Finally, using repeated measures MANOVA, we searched whether there was a difference in resting-state organization with the occurrence of 'dissociations' depending on 'language organization' (restricted to two factors ATYP and TYP, as the TYP_STRONG and TYP_MILD groups that were not different for Rs_DC and Rs_mIHHC were merged) by comparing their mean Rs_DC values and asymmetries (including a main effect of 'side' in the MANOVA), and SENT_CORE Rs_mIHHC.

All post hoc analyses were conducted using Tukey's HSD test for multiple comparisons.

### Anatomical variables

To investigate the brain structural differences in groups with different functional organization of language lateralization, we compared *corpus callosum* volume (CCvol) and asymmetries (left *minus* right) in gray matter (GMasym) and white matter (WMasym) hemispheric volumes. In this analysis, 'CROSSED' or 'CONGRUENT' was studied in interaction with the 'language organization' main effect restricted to two factors ('TYP' and 'ATYP').

First, to take into account variables that were found to covary with GMasym, WMasym and CCvol, we computed the residuals of MANCOVAs that included age, sex, total brain volume and handedness. These residuals of GMasym and WMasym were then entered in repeated measures ANOVA including a 'language organization' main effect restricted to two factors ('TYP' and 'ATYP') and dissociation ('CROSSED' or 'CONGRUENT') and their interaction as fixed factors and their interaction with the anatomical compartment (gray matter or white matter).

The residuals of CCvol were entered in ANOVA searching for an effect of a 'language organization' main effect restricted to two factors ('TYP' and 'ATYP'), an effect of dissociation with two factors ('CROSSED' or 'CONGRUENT') and their interaction.

## Cognitive variables

First, we performed a multiple linear regression analysis of the scores of the 11 tests of the cognitive battery, including sex, manual preference, age, education level and total intracranial volume as predictors since these variables have been shown to partly explain the variance in these scores (*Mellet et al., 2014a*). Residuals of the 11 regression analyses were then entered into PCA with a promax rotation. We used the *scree* criterion to determine the number of components to be retained.

The 'language organization' groups were compared with regard to their cognitive abilities through repeated measures MANCOVA including the four components of the PCA obtained from the residuals of the 11 scores. Finally, an impact of 'dissociation' on cognitive abilities was also tested in the ATYP and TYP_MILD groups (only two dissociations in TYP_STRONG).

Post hoc analyses were conducted using uncorrected Student's t-tests.

## Comparison of the different classifications for language lateralization

We also compared the present classification based on a multitask and multimodal approach to two other classifications that were previously applied to the same group of individuals, namely, the Gaussian mixture modeling classification on the HFLI obtained with the PROD$_{SENT\_WORD}$ contrast (*Mazoyer et al., 2014*) and an support vector machine approach applied at the voxel level, allowing us to classify the dominant and nondominant hemispheres of each participant according to their spatial pattern of activation during PROD$_{SENT\_WORD}$ (*Zago et al., 2017*).

To compare these three different classifications obtained in the 287 subjects, we used the '*ggalluvial*' R library to make an alluvial plot (*Brunson, 2020*). The alluvial plot allowed us to visualize, for each subject, their classification as TYP_STRONG TYP_MILD or ATYP issued from the present work, as typical, ambilateral (AMB), or strong-atypical (SA) based on HFLI (*Mazoyer et al., 2014*), and the classification of each of the hemispheres as dominant or nondominant obtained with support vector machine (*Zago et al., 2017*). Two plots were made, which included one for right-handed people and another for left-handers.

## Acknowledgements

The authors thank their colleagues Laure Zago and Emmanuel Mellet for their careful reading of the manuscript and Violaine Veraccia for expert technical assistance, supported by Ginesislab, a joint (University Bordeaux/CEA/CNRS and Fealinx) laboratory supported by the French government agency (ANR 16-LCV2-0006-01).

## Additional information

### Funding

| Funder | Grant reference number | Author |
| --- | --- | --- |
| Agence Nationale de la Recherche | ANR 16-LCV2-0006-01 | Marc Joliot |

The funders had no role in study design, data collection and interpretation, or the decision to submit the work for publication.

### Author contributions

Loïc Labache, Conceptualization, Data curation, Software, Formal analysis, Visualization, Methodology, Writing - original draft, Writing - review and editing; Bernard Mazoyer, Conceptualization, Resources, Formal analysis, Supervision, Methodology, Writing - original draft, Writing - review and editing; Marc Joliot, Data curation, Funding acquisition, Methodology, Writing - review and editing; Fabrice Crivello, Data curation, Visualization, Methodology, Writing - review and editing; Isabelle Hesling, Writing - original draft, Writing - review and editing; Nathalie Tzourio-Mazoyer, Conceptualization, Data curation, Formal analysis, Supervision, Validation, Methodology, Writing - original draft, Project administration, Writing - review and editing

## Author ORCIDs

Loïc Labache (iD) https://orcid.org/0000-0002-5733-0743
Bernard Mazoyer (iD) https://orcid.org/0000-0003-0970-2837
Marc Joliot (iD) http://orcid.org/0000-0001-7792-308X
Fabrice Crivello (iD) https://orcid.org/0000-0001-6950-984X
Isabelle Hesling (iD) https://orcid.org/0000-0002-3719-983X
Nathalie Tzourio-Mazoyer (iD) https://orcid.org/0000-0002-6236-4390

## Ethics

Human subjects: The Comité pour la Protection des Personnes dans la Recherche Biomédicale de Basse-Normandie approved the study protocol. All participants gave their informed, written consent, and received an allowance for their participation.

## Decision letter and Author response

Decision letter https://doi.org/10.7554/eLife.58722.sa1
Author response https://doi.org/10.7554/eLife.58722.sa2

## Additional files

### Supplementary files
• Transparent reporting form

### Data availability

All data generated or analysed during this study are included in the manuscript and supporting files. Source data files have been provided for all figures and tables.

The following dataset was generated:

| Author(s) | Year | Dataset title | Dataset URL | Database and Identifier |
|---|---|---|---|---|
| Labache, Mazoyer, Joliot, Crivello, Tzourio-Mazoyer | 2020 | BIL&GIN Sentence and Rest asymmetries - eLife | https://doi.org/10.5061/dryad.ht76hdrcf | Dryad Digital Repository, 10.5061/dryad.ht76hdrcf |

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
