## [Decision Letter]

**Acceptance summary:**

Characterizing language lateralization in the brain on an individual basis is crucial, particularly for functional mapping in neurosurgical patients, but a sensitive and specific method has been elusive. The present study offers novel insight into this old problem. This work will serve as an important reference point for future studies involving both basic and clinical research, as it advances our understanding of the patterns of language organization in the brain.

**Decision letter after peer review:**

Thank you for submitting your article "Typical and ATYP language brain organization based on intrinsic connectivity and multitask functional asymmetries" for consideration by *eLife*. Your article has been reviewed by two peer reviewers, and the evaluation has been overseen by a Reviewing Editor and Christian Büchel as the Senior Editor. The following individual involved in review of your submission has agreed to reveal their identity: Monika Polczynska (Reviewer #3).

The reviewers have discussed the reviews with one another and the Reviewing Editor has drafted this decision to help you prepare a revised submission.

The editors have judged that your manuscript is of interest, but (as described below) substantial revisions are required before a final decision can be made.

We would like to draw your attention to changes in our revision policy that we have made in response to COVID-19 (https://elifesciences.org/articles/57162). First, because many researchers have temporarily lost access to the labs, we will give authors as much time as they need to submit revised manuscripts. We are also offering, if you choose, to post the manuscript to bioRxiv (if it is not already there) along with this decision letter and a formal designation that the manuscript is "in revision at *eLife*". Please let us know if you would like to pursue this option. (If your work is more suitable for medRxiv, you will need to post the preprint yourself, as the mechanisms for us to do so are still in development.)

Summary:

Characterizing language lateralization in the brain on an individual basis is crucial, particularly for functional mapping in neurosurgical patients, but a sensitive and specific method has been elusive. The present study offers novel insight into this old problem, by examining a large sample of healthy volunteers, including over 50% LH, using several fMRI paradigms as well as structural MRI, and conducting exhaustive data analyses. The primary finding is of three different phenotypes characterised by different proportions of left handedness, different language-task based asymmetry, intra- and interhemispheric connectivity, and other measures including cognitive tasks. This work will serve as an important reference point for future studies involving both basic and clinical research, as it advances our understanding of the patterns of language organization in the brain.

Essential revisions:

1) A central claim made in the paper, to explain the results, is that "the dominant hemisphere may shift for different language functions in some individuals". This claim does not appear to be well supported and the authors are encouraged to consider an alternate, perhaps more parsimonious alternative which is that individuals rely to varying degrees on a variety of (non-linguistic) systems (with independent lateralization patterns) for different linguistic functions.

2) The finding of differences in the proportion of women between the language lateralization groups is consistent with prior findings of sex differences in language lateralization, which have in some instances been attributed to sex differences in the engagement of non-linguistic systems (e.g., Ullman, Nature Review Neuroscience, 2001; Kansaku et al., Cerebral Cortex, 2000; Kitazawa and Kansaku, Brain, 2005). Sex differences in the engagement of non-linguistic system may contribute to the differences in lateralization observed in the present study. For example, sex differences in visuospatial skills could contribute to the differences seen in this skill between the Typ_strong and Typ_mild groups (which have different proportions of women in the present study).

3) The various paradigms compared here may differ in their sensitivity and/or specificity for language mapping, which may lead to inconsistencies between them in activation lateralization. For example, the resting-state paradigm has been suggested to yield more bilateral language activation than explicit language tasks (Rolinski et al., Human Brain Mapping, 2020). Thus, dissociations in lateralization between paradigms may be explained (at least in part) by technical differences in the paradigms, rather than physiological shifts in lateralization for different language functions.

4) Provide additional explanation and justification for the classification into three groups, since this is central to the paper. The classification seems reasonable and yields interesting results. However, the three clusters do seem to overlap substantially in features (at the boundaries), bringing into question whether the data are best characterized as discrete categories or as a continuous distribution.

---

## [Author Response]

Essential revisions:1) A central claim made in the paper, to explain the results, is that "the dominant hemisphere may shift for different language functions in some individuals". This claim does not appear to be well supported and the authors are encouraged to consider an alternate, perhaps more parsimonious alternative which is that individuals rely to varying degrees on a variety of (non-linguistic) systems (with independent lateralization patterns) for different linguistic functions.

We thank the reviewers for this comment. First, we would like to apologize if the way we phrased our results leads the reader to the conclusion that the existence of dissociations was the central claim of the paper. Notwithstanding their importance, such dissociations concerned only 23 participants (8%) of the sample. Strikingly, while dissociations happened in 40% of atypicals, they were only present in 4% of “typicals” (merging TYP_mild and TYP_strong), i.e. 10 times less frequent. So, the central claim of our paper is actually that co-lateralization of the 3 language functions is the rule, whereas their dissociation is rare in healthy volunteers, although they exist.

To illustFigure 8 rate that co-variation of lateralization is the rule, we plotted the between-tasks correlation graphs of BOLD signal asymmetries, (typicals are in green and atypicals in red) for the reviewer information (Author response image 1).

**Author response image 1. sa2fig1:** 

Note also that we ensured that the lateralization measurements across tasks were comparable by setting a paradigm with close conditions across tasks and with the aim of limiting the potential differences in the engagement of attention and executive functions (that could lead to the right hemisphere involvement):– the tasks were very similar, involving single sentence processing whatever the modality;

– the control task we designed involves word-lists which is a high order language task that also calls for intentional processes;

– asymmetries were not calculated on the whole hemisphere, rather we targeted the areas involved in sentence processing.

It seems thus unlikely that some individuals shifted their asymmetry because of higher attention control for one language component relative to the others.

Actually, it has been shown that asymmetries measured during language tasks compared to a high reference condition (see Binder for example) are an appropriate marker of language dominance (see Dym). In the clinical literature, but also in healthy volunteers’ studies, dissociations have been previously observed. We thus believe that the change in dominance is the most likely hypothesis for the change in lateralization we observed in case of dissociation. As developed in the manuscript, the fact that it is more frequent in atypicals is also consistent with the fact that we have shown that their intrinsic connectivity at rest attests for a bilateral organization of language network and a strong inter-hemispheric correlation. Indeed, concordance of asymmetries in the 3 tasks is the rule as shown in Author response image 1.

Following the advice of the reviewer we have simplified the Abstract of the revised version of the manuscript (see below) and we hope it makes the major points of the results of the present work clearer, including the observation that while colateralization across tasks is the rule, dissociations are rare and mainly observed in ATYP individuals.

2) The finding of differences in the proportion of women between the language lateralization groups is consistent with prior findings of sex differences in language lateralization, which have in some instances been attributed to sex differences in the engagement of non-linguistic systems (e.g., Ullman, Nature Review Neuroscience, 2001; Kansaku et al., Cerebral Cortex, 2000; Kitazawa and Kansaku, Brain, 2005). Sex differences in the engagement of non-linguistic system may contribute to the differences in lateralization observed in the present study. For example, sex differences in visuospatial skills could contribute to the differences seen in this skill between the Typ_strong and Typ_mild groups (which have different proportions of women in the present study).

The reviewer is right as regards the existence of sex differences on visuospatial skills. However, there is no evidence in the literature of an association between asymmetries of activation during language tasks and visuo-spatial performance in TYP individuals. Moreover, we have shown in Mellet et al., 2014, that individuals with low hemispheric lateralization during language production have lower performance in all domains (verbal and visuo-spatial), whereas in the present study, a sex-related difference in lateralization is seen in TYP individuals with leftward lateralization.

In addition, we have previously reported (Beaucousin et al., 2011) that men tend to recruit more attentional and executive areas of the right hemisphere when processing emotional sentences. Such a larger involvement of attentional areas during sentence processing would lead to lower leftward asymmetries in men than in women (note that this reasoning only works if considering global hemispheric asymmetries, while in the present paper we targeted sentence areas).

Overall, this is why we believe that we observed here a genuine lower leftward lateralization in TYP_mild that, as mentioned by the reviewer, is associated with a higher proportion of women in that group. In other words, we report that in typicals, women are more frequently of the mild type while men are more frequently strongly lateralized, although we have no data in the present study allowing us to interpret or find a potential cause to this observation.

3) The various paradigms compared here may differ in their sensitivity and/or specificity for language mapping, which may lead to inconsistencies between them in activation lateralization. For example, the resting-state paradigm has been suggested to yield more bilateral language activation than explicit language tasks (Rolinski et al., Human Brain Mapping, 2020). Thus, dissociations in lateralization between paradigms may be explained (at least in part) by technical differences in the paradigms, rather than physiological shifts in lateralization for different language functions.

The reviewer is right when mentioning that the resting-state is less lateralized and it is actually what is shown in the present work where ATYP individuals, although having a rightward lateralization during the language tasks, have symmetrical DC at rest in the sentence core network.

However, in our study, the definition of dissociation was based only on the task-induced asymmetries measured during the 3 language tasks, and did not concern resting state. Accordingly, there was no between-paradigms element in this observation. As developed in the answer to point 1, the 3 language tasks were elaborated to conduct such comparisons since they had the same structure, were constituted of equivalent tasks, and of equivalent reference tasks.

In addition, although the phenomenon of dissociations is of interest, recall that it concerns only a fraction of 23 participants which represents 8% of the sample, including 40% of individuals with ATYP organization.

4) Provide additional explanation and justification for the classification into three groups, since this is central to the paper. The classification seems reasonable and yields interesting results. However, the three clusters do seem to overlap substantially in features (at the boundaries), bringing into question whether the data are best characterized as discrete categories or as a continuous distribution.

We agree with the reviewer and added the following explanation and justification for the classification at the beginning of the “Statistical Analysis” section:

“In a previous work (Mazoyer et al., 2014), we have shown that the distribution of lateralization for sentence production, although of continuous nature, could be used to classify individuals into 3 discrete categories. […] Rather, the optimal number of clusters for this multivariate classification was obtained using a fully unsupervised methodology and a combination of 30 statistical criteria.”

That an unsupervised classifier succeeded at characterizing this multivariate dataset in 3 discrete categories only constitutes by itself a validation of our approach. This being said, as pointed out by the reviewers, classification methods based on multiple continuous variables almost inevitably lead to clusters that overlap at their boundaries for some of these variables. Such overlaps do not invalidate the classification approach. Rather they reflect the importance of simultaneously accounting for all variables, as opposed to classifying by looking at each variable separately.